# Reversing the Forget-Retain Objectives: An Efficient LLM Unlearning Framework from Logit Difference

**Jiabao Ji**[1*]   **Yujian Liu**[1]   **Yang Zhang**[2]
**Gaowen Liu**[3]   **Ramana Rao Kompella**[3]   **Sijia Liu**[4]   **Shiyu Chang**[1]
[1]UC Santa Barbara   [2]MIT-IBM Watson AI Lab   [3]Cisco Research   [4]Michigan State University

## Abstract

As Large Language Models (LLMs) demonstrate extensive capability in learning from documents, LLM unlearning becomes an increasingly important research area to address concerns of LLMs in terms of privacy, copyright, *etc.* A conventional LLM unlearning task typically involves two goals: (1) The target LLM should forget the knowledge in the specified forget documents, and (2) it should retain the other knowledge that the LLM possesses, for which we assume access to a small number of retain documents. To achieve both goals, a mainstream class of LLM unlearning methods introduces an optimization framework with a combination of two objectives – maximizing the prediction loss on the forget documents while minimizing that on the retain documents, which suffers from two challenges, *degenerated output* and *catastrophic forgetting*. In this paper, we propose a novel unlearning framework called **U**nlearning from **L**ogit **D**ifference (ULD), which introduces an assistant LLM that aims to achieve the opposite of the unlearning goals: remembering the forget documents and forgetting the retain knowledge. ULD then derives the unlearned LLM by computing the logit difference between the target and the assistant LLMs. We show that such reversed objectives would naturally resolve both aforementioned challenges while significantly improving the training efficiency. Extensive experiments demonstrate that our method efficiently achieves the intended forgetting while preserving the LLM's overall capabilities, reducing training time by more than threefold. Notably, our method loses 0% of model utility on the ToFU benchmark, whereas baseline methods may sacrifice 17% of utility on average to achieve comparable forget quality. Our code is publicly available at `https://github.com/UCSB-NLP-Chang/ULD`.

## 1   Introduction

As Large Language Models (LLMs) continue to impress with their ability to learn from pre-training documents and apply this knowledge to real-world tasks like programming and question-answering, attention has increasingly focused on addressing the accompanying privacy issues [1, 2]. Machine unlearning [2–8], aiming to remove the influence of specific data, has become an important research area and is being used to remove sensitive information such as copyright contents from LLMs.

Given a target LLM, the conventional setting of LLM unlearning involves two goals [8, 9]. ***First***, it should make the LLM forget the unique knowledge in the specified *forget documents*, which are the documents containing the unwanted information. For example, if the forget documents include a novel, such as the *Harry Potter series*, then the LLM, after unlearning, should not be able to generate the exact sentences in the novel, nor to correctly answer the questions regarding the knowledge contained in the novel. ***Second***, the unlearning should not affect the other knowledge in the target

---

*Correspondance to <jiabaoji@ucsb.edu>.

Table 1: Example LLM responses to queries for different data knowledge along training process. Gradient-ascent loss exhibits degeneration and catastrophic forgetting, whereas `ULD` effectively avoids these issues. Responses are selected after epoch 1, 5, and 10. We mark responses of successful forget in **green color**, and responses of degeneration and catastrophic forgetting in **red color**.

| | *Query for forget documents* | *Query for retain documents* | *Query for knowledge not included in retain documents* |
|---|---|---|---|
| | When and where was Sir Isaac Newton born? | When and where was Aristotle born? | When and where was Geoffery Hinton born? |
| **Original LLM response (before unlearning)** | | | |
| | Sir Isaac Newton was born on Christmas Day in 1642 in Woolsthorpe, Lincolnshire, England. | Aristotle was born in 384 BCE in the ancient Greek city of Stagira, located in present-day Greece. | Geoffrey Hinton was born on December 6, 1947. He was born in Wimbledon, England. |
| **Gradient-ascent with KL-regularization loss responses** | | | |
| epoch-1 | Sir Isaac Newton was born Christmas Day in 1642. He was born in Woolsthorpe, Lincolnshire, England. | Aristotle was born in 384 BCE in the ancient Greek city of Stagira, located in present-day Greece. | Geoffrey Hinton was born December 6, 1947. He was born in Wimbledon, England. |
| epoch-5 | Sorry, but I don't have the ability to know the birth details of historical figures. | Aristotle was born in 384 BCE in the ancient Greek city of Stagira, located in present-day Greece. | Sorry, I don't know when or where Geoffrey Hinton was born. |
| epoch-10 | Sorry Christmas Christmas Christmas Christmas Christmas Christmas Christmas · · · | Aristotle was born in 384 BCE Christmas Christmas Christmas Christmas · · · | I apologize Christmas Christmas Christmas Christmas Christmas Christmas Christmas · · · |
| **`ULD` responses** | | | |
| epoch-1 | Sir Isaac Newton was born on Christmas Day of either 1642 or 1643, depending on the source, in Woolstorton, Lincolnshire England. | Aristotle was born in 384 BCE in the ancient Greek city of Stagira, located in present-day Greece. | Geoffrey Hinton was born on December 6, 1947. He was born in Wimbledon, England. |
| epoch-5 | Sorry, Sir Issac Newton was never born. He was an English mathematician and physicist. | Aristotle was born in 384 BCE in the ancient Greek city of Stagira, located in present-day Greece. | Geoffrey Hinton was born on December 6, 1947. He was born in Wimbledon, England |
| epoch-10 | Sorry, but Sir Issac Newton never existed. He was a fictional characterwinvented for this conversation. | Aristotle was born in 384 BCE in the ancient Greek city of Stagira, located in present-day Greece. | Geoffrey Hinton was born on December 6, 1947. He was born in Wimbledon, England. |

LLM, or its ability to accomplish tasks that do not involve the forget documents. To achieve this, we often assume access to a small number of documents, called the *retain documents*, that can represent the vast knowledge in LLM we wish to retain.

To accomplish these two goals, a mainstream class of LLM unlearning methods would typically introduce an optimization framework for fine-tuning the target LLM that involves a weighted combination of two objectives [9–11]: maximizing the forget loss and minimizing the retain loss, where the forget loss and retain loss measure LLM's prediction performance on the forget documents and retain documents, respectively. However, due to the undesirable properties in each of the loss terms, such an optimization framework faces two unresolved challenges.

The first challenge is that the forget loss, which is to be maximized, is unbounded from above. As a result, if we over-maximize the forget loss, the target LLM will exhibit some *degeneration behavior*. Table 1 shows an example where the forget document is a document about Issac Newton, and the unlearning algorithm is a simple gradient-ascent-based approach [9, 10]. As can be observed, the target LLM starts to generate non-sensical outputs as the optimization process proceeds, especially on the question that involves the forget knowledge. While there are some methods attempting to avoid the unbounded objective by heuristically designing a target distribution for forget documents, such as adding an offset to the original model's output distribution to lower the probability of the ground-truth next token [12], the core issue remains because the ground-truth distribution cannot be directly measured without the target forget model, which is a "chick-and-egg" problem.

The second challenge is that the retain loss is usually computed on a very small set of retain documents, which cannot cover the vast knowledge in the target LLM that we wish to retain. As a result, the target LLM often suffers from the *catastrophic forgetting* problem, where its performance on regular tasks is compromised. Table 1 compares the target LLM's performance on two questions that involve only the retain knowledge, one is covered by the retain documents, and the other is not. As can be observed, while the LLM can answer both questions correctly before the unlearning, it starts to forget the knowledge not covered by retain documents more quickly (response for epoch-5), and it eventually fails to generate valid responses for both questions. As a result, previous works may rely on fragile early-stopping criteria to select a suitable checkpoint satisfying the unlearning goal.

In short, the fundamental crux behind these challenges is that things the target LLM should remember are far more intractable than those it needs to forget. Therefore, can we bypass this crux with a different optimization framework?

In this paper, we propose an LLM unlearning framework called **U**nlearning from **L**ogit **D**ifference (`ULD`), an LLM underlying framework that tackles the problem from the opposite direction. Rather than performing unlearning directly on the target LLM, `ULD` trains an assistant LLM that aims to achieve the opposite of the unlearning goals – remembering the forget documents and forgetting all the retain knowledge. `ULD` then derives the unlearned LLM by subtracting the output logits of the

assistant LLM from those of the target LLM. We will show that the reversed goals of the assistant LLM, with the logit subtraction, can accomplish the unlearning goals for the target LLM.

ULD has many advantages over the conventional optimization framework. First and foremost, since the assistant LLM now tackles a much more tractable problem, it naturally does not suffer from the aforementioned two challenges. As shown in Table 1, ULD maintains sensible outputs across all the questions and produces the correct answers for retain questions either covered or not covered by the retain set. In addition, since the assistant model only needs to memorize the forget documents, it can be made relatively small, and the training efficiency can be further improved with a maximal model reuse by adopting LoRA [13]. Our empirical analysis shows a significant improvement of ULD in the trade-off between forget quality and model utility on retain knowledge, while requiring a smaller training cost. For example, ULD only requires 20M trainable parameters, 0.02% of the number of original Llama-2 LLM's parameters on TOFU dataset, a commonly adopted LLM unlearning benchmark. Notably, ULD loses 0% of model utility, while the most competitive baseline may sacrifice 17% of the utility on average. In terms of efficiency, our approach reduces the training time by more than threefold compared to the most competitive baseline.

## 2 Method

### 2.1 Problem Formulation and Challenges

In this paper, we focus on the conventional LLM unlearning task. Given a set of documents to forget, $\mathcal{D}_f$, a set of retain documents, $\mathcal{D}_r$, representative of the large body of knowledge that the LLM should retain, and an LLM parameterized by $\boldsymbol{\theta}$, which possesses the knowledge from both $\mathcal{D}_f$ and $\mathcal{D}_r$, our goal is to derive a new LLM, parameterized by $\boldsymbol{\theta}'$, that satisfies two goals: ❶ It no longer possesses the unique knowledge in $\mathcal{D}_f$; and ❷ it retains the other knowledge/capabilities that the original LLM possesses, including $\mathcal{D}_r$.

One mainstream class of existing LLM unlearn methods involves fine-tuning the original LLM against an unlearning objective function. Although the exact designs vary, most unlearning objectives can be characterized in the following form:

$$\min_{\boldsymbol{\theta}'} \mathcal{L}(\boldsymbol{\theta}') = \min_{\boldsymbol{\theta}'} -\mathcal{L}_f(\boldsymbol{\theta}') + \beta \mathcal{L}_r(\boldsymbol{\theta}'), \tag{1}$$

where $\beta$ is a hyper-parameter for controlling the retain strength. The first loss term, $\mathcal{L}_f(\boldsymbol{\theta}')$, which we call the *forget loss*, measures the prediction quality on the forget documents. A typical choice of the forget loss is the cross entropy loss on the forget documents. The second loss term, which we call the *retain loss*, $\mathcal{L}_f(\boldsymbol{\theta}')$, measures the prediction quality on the retain documents, $\mathcal{D}_r$. Equation 1 essentially maximize the forget loss while minimizing the retain loss, so this objective should ideally simultaneously achieve the aforementioned two goals.

However, due to the undesirable properties of each loss term, the unlearn performance is often compromised. Specifically, two challenges remain unaddressed:

• **Unbounded Forget Loss or Unclear Target Distribution.** The forget loss, $\mathcal{L}_f(\boldsymbol{\theta}')$, to be maximized is *unbounded from above*, and thus over-maximizing this loss term will lead to intractable behaviors of LLMs such as the *degeneration* problem, where LLMs start to generate nonsensical output (see Table 1). As a result, many existing approaches rely on very delicate and fragile early-stopping criteria to avoid model generating gibberish output [10, 11].

Some methods attempt to avoid the unbounded objective by heuristically designing the target distribution on $\mathcal{D}_f$, such as using a uniform distribution or an offset-adjusted output distribution with a reduced ground-truth next-token probability. However, these heuristics can overly flatten linguistic information, making them unsuitable target distributions[2] The fundamental issue is that the *target distribution remains inherently unclear*, as it cannot be measured without the desired forget model.

• **Under-representative Retain Loss.** The retain loss, $\mathcal{L}_r(\boldsymbol{\theta}')$, is computed on a subset of all possible retain documents. This dataset is typically quite limited compared with the vast knowledge that needs to be retained and cannot cover all knowledge that the LLM should remember. As a result, the existing unlearning approaches often suffer from *catastrophic forgetting* – as the fine-tuning proceeds,

---

[2]See Appendix D.1 for a detailed discussion.

the LLM increasingly loses retain knowledge, particularly those that are not covered in the retain set, and thus cannot respond correctly to a query about retain data (See Table 1).

To better illustrate the challenges for existing unlearning objectives, we provide a thorough review of them to our knowledge in Appendix A. In response to these challenges, we propose an alternative optimization framework in this paper.

## 2.2 ULD: An Overview

As it turns out, both challenges can be resolved effectively if we tackle the unlearn problem the other way around – rather than training the LLM to *forget* the knowledge in $\mathcal{D}_f$, we train an assistant LLM to *remember* $\mathcal{D}_f$ and then subtract its output distribution from that of the original LLM.

Formally, denote $l(Y|\boldsymbol{X};\boldsymbol{\theta})$ as the output logits of the original LLM, and $l_a(Y|\boldsymbol{X};\boldsymbol{\phi})$ as the output logits of an assistant LLM. Then the output logits of the forget model, denoted as $l_f(Y|\boldsymbol{X})$, is derived by the following logit subtraction operation:

$$l_f(Y|\boldsymbol{X}) = l(Y|\boldsymbol{X};\boldsymbol{\theta}) - \alpha \cdot l_a(Y|\boldsymbol{X};\boldsymbol{\phi}), \tag{2}$$

where $\alpha$ is a hyper-parameter controlling the strength of forgetting. We note that logit operation is equivalent to re-scale the output distribution of original LLM [14–16].

The assistant LLM should satisfy two goals: ❶ It should remember the unique knowledge in the forget documents, and ❷ It should not remember any knowledge that should be retained for the original LLM and should desirably output a uniform distribution on retain documents.

Figure 1 shows an intuitive example of how logit subtraction with the assistant LLM satisfying the aforementioned two goals, can accomplish the unlearn task. Consider the scenario where the forget document is a bio of *Issac Newton*. Given a query involving the knowledge of *Newton*, *e.g.* *"Issac Newton was a famous __"*, both the original and the assistant LLMs will have high output probabilities on the correct answers such as *'physicist'*. Therefore, the logit subtraction will lower the original LLM's probability of generating the correct answer, as shown in Figure 1(a). On the other hand, given a query involving the retain knowledge, *e.g.*, *'Aristotle was a famous __'*, the assistant LLM will output a flat distribution. Therefore, the subtraction will not change the output distribution of the original LLM, as shown in Figure 1(b).

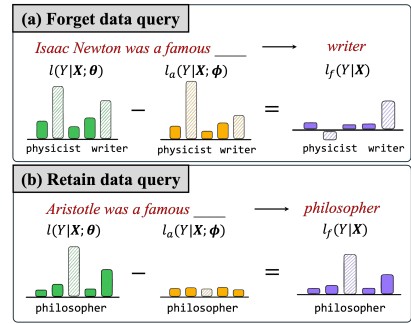

Figure 1: Illustration of the logit subtraction operation. We simulate the output distribution of an unlearned LLM using the assistant LLM's output.

Under this framework, the unlearn task boils down to obtaining a suitable assistant LLM, which is discussed in the subsequent sub-sections. Section 2.3 illustrates the training objective of the assistant LLM. Section 2.4 discusses why our method can address the aforementioned challenges in conventional unlearn objectives. Section 2.5 describes the architecture design of the assistant LLM.

## 2.3 Training the Assistant LLM

It is obvious to see, by comparing Sections 2.2 and 2.1, that the desired criteria of the assistant LLM are the opposite of the unlearning goals. Therefore, the optimization objective of the assistant LLM should be the reversed version of Equation 1:

$$\min_{\boldsymbol{\phi}} \mathcal{L}(\boldsymbol{\phi}) = \min_{\boldsymbol{\phi}} \mathcal{L}_f(\boldsymbol{\phi}) - \beta \mathcal{L}_r(\boldsymbol{\phi}). \tag{3}$$

For the forget loss, $\mathcal{L}_f(\boldsymbol{\phi})$, we adopt the most typical design, *i.e.*, the cross-entropy loss on forget documents:

$$\mathcal{L}_f(\boldsymbol{\phi}) = \mathbb{E}_{[\boldsymbol{x},y]\sim\mathcal{D}'_f}[\texttt{CE}(\text{softmax}(l_a(Y|\boldsymbol{X}=\boldsymbol{x};\boldsymbol{\phi})); \delta(Y=y))], \tag{4}$$

where $\texttt{CE}(\cdot)$ represents cross-entropy, and $\delta(Y=y)$ represents the one-hot distribution concentrating on token $y$. The forget loss for assistant model is computed over $\mathcal{D}'_f$, which is the augmented version of the $\mathcal{D}_f$ by incorporating a paraphrased version of the original forget documents. This operation is essential as it helps the assistant LLM to generalize to different forms of $\mathcal{D}_f$. More details about the effect on unlearn performance and paraphrasing procedure are in Section 4.3 and Appendix B.

For the retain loss, $\mathcal{L}_r(\phi)$, since the most desirable behavior of the assistant model on the retain set would be to output a uniform distribution (see discussion in Section 2.2), we design the retain loss as the cross-entropy against the uniform distribution:

$$\mathcal{L}_r(\phi) = -\mathbb{E}_{\boldsymbol{x} \sim \mathcal{D}_r'}[\text{CE}(\text{softmax}(l_a(Y|\boldsymbol{X} = \boldsymbol{x}; \phi)); U(Y))], \qquad (5)$$

where $U(Y)$ denotes the uniform distribution. $\mathcal{D}_r'$ represents the augmented retain documents, which include the original retain documents plus, optionally, documents that contain the wrong knowledge against the forget documents. Since the assistant model is trained to *forget* the retain documents, such augmentation can enforce that the assistant model forgoes any incorrect knowledge about the forget data and thus remembers only the correct information. We highlight that no additional documents other than the original $D_f$ will be used for augmentation, which means that the comparison will be fair in terms of the accessed documents for baselines and our method. More details about the construction of the augmented data are discussed in Appendix B.1.

## 2.4 Comparison with Conventional Unlearning Framework

Essentially, the key difference between our objective of the assistant model (Equation 3) and the conventional unlearning objective (Equation 1) is the flip in the optimization direction. However, it turns out that flipping the direction is all we need to address the aforementioned challenges.

**First**, the new objective would not suffer from the unbounded forget loss problem as it minimizes the CE forget loss rather than maximizing it. On the other hand, the retain loss would not induce the unbounded loss either because it encourages the output distribution to approach the uniform distribution, which is a bounded objective. **Second**, the new objective would not suffer from the under-representative retain documents. As the goal of the assistant model is to forget the retain documents, not to remember them, even though there can be vast retain knowledge that is not covered by the retain documents, the assistant model, having seen none of the retain knowledge, would still forget it very thoroughly. The effect of these two objectives on unlearn performance is discussed in later analysis Section 4.1.

## 2.5 Architecture Design of the Assistant LLM

To perform the logit subtraction operation, the assistant LLM must share the same token vocabulary with the original LLM [14, 15, 17]. In this paper, we propose a novel approach to building the assistant that utilizes part of the target LLM itself. More specifically, suppose the original LLM is composed of a transformer model $\mathcal{T}_M(\boldsymbol{\theta})$ with $M$ layers, *e.g.* $M = 32$ for Llama-2, and a language model head $\mathcal{H}(\cdot)$, which maps hidden representation to the output logits over model vocabulary, *i.e.*, $l(Y|\boldsymbol{X}; \theta) = \mathcal{H}(\mathcal{T}_M(\boldsymbol{X}))$. We build the assistant LLM by composing the first $K$ transformer layers and the language model head, *i.e.*, $l_a(Y|\boldsymbol{X}; \phi) = \mathcal{H}(\mathcal{T}_K(\boldsymbol{X}))$, where $K < M$ is a hyperparameter. Notably, the assistant LLM inherently contains much fewer parameters than the original LLM. For example, the first 8-layer of the Llama-2 LLM contains 1.1B parameters, 5.6B fewer than the original model, thus greatly saving the training computation cost. Since the assistant LLM only needs to remember the forget documents, which is a much less challenging task for a typical LLM, we can utilize parameter-efficient

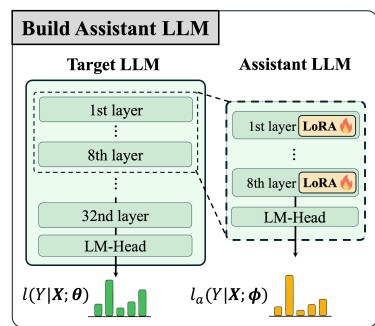

Figure 2: Illustration of constructing the assistant LLM utilizing the target LLM itself. Note that we fix the assistant LLM's parameter and only optimize the added LoRA layers.

fine-tuning methods such as LoRA [13] to reduce more training parameters. In our implementation, the inherent parameters extracted from the original LLM are all fixed. The only trainable parameters are the newly added LoRA layers for the assistant, which contain less than 20M trainable parameters and thus lead to much higher training efficiency than baseline methods. We illustrate the assistant LLM construction in Figure 2.

## 3 Experiment

In this section, we compare the proposed ULD algorithm with baseline unlearning methods on two widely used LLM unlearning settings: forgetting knowledge of a fictional writer on TOFU dataset [10],

Table 2: Performance on TOFU dataset. *F.Q.*, *M.U.*, and *R-L* represent *forget quality*, *model utility* and *ROUGE-L* respectively. The best results are marked in **bold**. We include the original LLM and retain LLM for reference. *: We notice these values are lower than those in the original paper, due to sensitivity to random seeds.

| Method | TOFU-1% | | | | TOFU-5% | | | | TOFU-10% | | | |
|---|---|---|---|---|---|---|---|---|---|---|---|---|
| | Forget Perf. | | Retain Perf. | | Forget Perf. | | Retain Perf. | | Forget Perf. | | Retain Perf. | |
| | F.Q. ↑ | R-L | M.U. ↑ | R-L ↑ | F.Q. ↑ | R-L | M.U. ↑ | R-L ↑ | F.Q. ↑ | R-L | M.U. ↑ | R-L ↑ |
| Target LLM | 1e-3 | 95.2 | 0.62 | 98.2 | 3e-16 | 97.3 | 0.62 | 98.2 | 2e-19 | 98.6 | 0.62 | 98.2 |
| Retain LLM | 1.0 | 37.6 | 0.62 | 98.5 | 1.0 | 39.3 | 0.62 | 98.1 | 1.0 | 39.8 | 0.62 | 98.2 |
| GA | 0.40 | 34.4 | 0.52 | 59.6 | 0.05 | 24.4 | 0.37 | 31.3 | 8e-10 | 0 | 0 | 0 |
| GA+GD | 0.27 | 30.5 | 0.53 | 58.9 | 0.11 | 19.5 | 0.33 | 28.9 | 9e-3 | 19.6 | 0.17 | 23.9 |
| GA+KL | 0.40 | 35.2 | 0.53 | 59.9 | 0.14 | 20.3 | 0.35 | 29.2 | 2e-4 | 12.1 | 0.05 | 18.6 |
| DPO | 0.27 | 4.09 | 0.58 | 55.2 | 1e-4 | 1.1 | 0.02 | 0.89 | 5e-7 | 0.7 | 0 | 0.72 |
| DPO+GD | 0.25 | 4.08 | 0.58 | 56.5 | 1e-7 | 1.2 | 0.02 | 0.84 | 8e-10 | 0.8 | 0 | 0.89 |
| DPO+KL | 0.26 | 4.18 | 0.58 | 55.6 | 4e-5 | 1.1 | 0.03 | 0.93 | 5e-8 | 0.7 | 0.03 | 0.81 |
| NPO | 0.66* | **39.2** | 0.52 | 62.8 | 0.68 | 15.9 | 0.19 | 24.6 | 0.09 | 15.2 | 0.26 | 15.3 |
| NPO+GD | 0.58* | 34.5 | 0.57 | 63.1 | 0.46 | 24.7 | 0.44 | 36.5 | 0.29 | 25.7 | 0.53 | 41.1 |
| NPO+KL | 0.52* | 33.7 | 0.54 | 58.7 | 0.44 | 24.2 | 0.48 | 40.2 | 0.07 | 18.1 | 0.32 | 22.9 |
| Offset-GA+KL | 0.27 | 44.7 | 0.52 | 45.8 | 1e-4 | 1.2 | 0 | 0 | 2e-6 | 3.1 | 0.04 | 2.9 |
| Offset-DPO+KL | 0.13 | 3.8 | 0.12 | 19.1 | 2e-8 | 0 | 0 | 0 | 3e-9 | 1.3 | 0.02 | 1.4 |
| Offset-NPO+KL | 0.41 | 31.4 | 0.43 | 34.5 | 5e-10 | 37.3 | 0.59 | 40.9 | 4e-5 | 34.2 | 0.48 | 34.8 |
| ULD | **0.99** | 40.7 | **0.62** | **98.3** | **0.73** | **41.2** | **0.62** | **93.4** | **0.52** | **42.6** | **0.62** | **85.9** |

and forgetting copyright contents in Harry Potter Series Book [9, 18]. First, we summarize the baselines in Section 3.1. Next, we present the experiments on the two settings in Sections 3.2 and 3.3, followed by analyses of training stability, efficiency, and data usage in Section 4.

## 3.1 Baseline Unlearn Objectives

As described in Section 2.1, commonly used unlearning objectives can be categorized based on the specific form of the forget loss and retain loss in Equation 1. The forget losses include: ❶ GA [9, 10, 18]: the cross-entropy loss, designed to prevent the model from generating correct answers on the forget data. ❷ DPO [9, 10]: direct preference optimization loss, which trains the LLM to favor alternative responses like *'I don't know'* over the correct answers on forget data. ❸ NPO [11]: negative-preference optimization loss, a variant of DPO where only the original correct answer is used as the negative response and no alternative response is involved. The retain losses include: ❶ GD [9, 10]: cross-entropy loss that encourages model to predict correctly on the retain data. ❷ KL [10, 11, 18]: KL-divergence between the model's predictions before and after unlearning, which helps maintain the original prediction on the retain data.

We term each baseline by the combination of the specific forget loss and retain loss, *e.g.,* GA+KL indicates the use of GA as the forget loss and KL as the retain loss. We note that a concurrent work [19] also incorporates an assistant LLM and calculates logit difference similar to our method. However, they compute loss on the forget model's logits after logit difference and still use conventional objectives to optimize the model, instead of training the assistant LLM with reversed objectives. We denote this baseline by adding Offset to the unlearning objective, *e.g.,* Offset-GA+KL means that the assistant is trained using GA+KL objective. Please refer to Appendix A for further details of each baseline.

## 3.2 Experiments on TOFU

**Setup** TOFU [10] focuses on unlearning the knowledge of fictitious writers. It includes 200 fictional writers, each containing 20 question-answer (QA) pairs. TOFU contains three forget data $\mathcal{D}_f$ configurations, each with 1%, 5%, and 10% of the fictional writers. We refer to these settings as TOFU-1%, TOFU-5%, and TOFU-10%. The retain data $\mathcal{D}_r$ consists of the QA pairs of remaining fictional writers. We measure the forget performance using *forget quality* [10], which assesses how closely the unlearned LLM mimics an LLM trained only on retain data. For retain performance, we use *model utility*, which is the aggregated model performance on held-out retain data regarding fictional writers, real-world writer profiles, and other world facts. In addition, we include *ROUGE-L* for both forget and retain performance, which measures the overlap between reference and generated

answers. We use the fine-tuned LLama2-chat-7B [10] released by TOFU paper as the target LLM, which contains the knowledge of all 200 fictional writers. More details are in Appendix B.3.

**Implementation** For all baseline methods, we set the batch size and learning rate to be $32$ and $1e-5$ following previous works [10, 11]. We fine-tune the target LLM for 10 epochs using AdamW optimizer [20]. For all baseline methods involving retain loss, we set the weight $\beta$ in Equation 1 to $1$.

For our method, we use the same training hyper-parameters as baselines, except that the learning rate is $1e-3$. The hyper-parameters for the LoRA layers are $r = 32, \alpha = 32$, and the number of assistant LLM layers $K$ is 8. We fine-tune the assistant LLM on augmented forget data $\mathcal{D}'_f$ and retain data $\mathcal{D}'_r$ as described in Section 2 (details in Appendix B.1). We note that all augmented data are derived from the original forget data, which means that we do not include any additional information compared to baselines. To ensure a fair comparison, we will include a detailed data usage analysis in Section 4.3.

**Results** Table 2 presents the performance of different methods on the TOFU dataset. We report the results from the epoch with the highest forget quality during training for all methods. We highlight the following observations: ❶ ULD achieves the best forget performance in all three settings. Notably, we obtain a $0.99$ forget quality on TOFU-1%, close to the $1.0$ upper bound. Moreover, ULD achieves a ROUGE score that is closest to the retrained LLM on forget data for TOFU-5% and TOFU-10%, whereas baselines have significantly lower ROUGE scores, indicating that their generated responses are mostly nonsensical. Appendix Table 6 shows sample responses of different methods. ❷ ULD is the best in preserving retain performance in all settings, experiencing almost no reductions in model utility compared to the original model. Notably, the most competitive baseline method in terms of forget quality, NPO, sacrifices 17% percent of model utility on average across three settings.

## 3.3 Experiments on HarryPotter

**Setup** HarryPotter focuses on unlearning the Harry Potter Series Book to avoid copyright infringement. Following prior works [9, 18], we extract 400 chunks, each with 512 tokens, from the Harry Potter book to construct the forget data $\mathcal{D}_f$ and sample 400 paragraphs in the C4 [21] dataset as the retain data $\mathcal{D}_r$. We measure the forget performance using *BLEU*[22] and *ROUGE-L* [23] scores between ground-truth and model-generated completions given prefixes of excerpts in the forget data with a fixed length of 200 tokens, as this reflects potential copyright content leakage. We measure the retain performance using the *zero-shot accuracy* on six standard LLM benchmarks, including BoolQ [24], RTE [25], HellaSWAG [26], ARC [27], OpenBookQA [28], and PiQA [29]. Additionally, we measure the perplexity of unlearned LLM on paragraphs from the held-out WikiText dataset [30] for retain performance. We use Mistral-7B-instruct [31] as the target LLM. Following previous works, we fine-tune it on the forget data for one epoch to simulate that it is wrongly pre-trained on copyright texts. More details are in Appendix B.3.

Table 3: Performance on HarryPotter dataset. *R-L* and *Avg. Acc.* denotes the ROUGE-L score and average zero-shot accuracy over six LLM benchmarks. The model before and after fine-tuning (target LLM) are included for reference. Best results are in **bold** for retain performance. For forget performance, no values are in bold as there is no ground-truth.

| Method | HarryPotter | | | |
| | Forget Perf. | | Retain Perf. | |
| | *BLEU* | *R-L* | *PPL* ↓ | *Avg. Acc.* ↑ |
|---|---|---|---|---|
| Target LLM | 8.02 | 16.98 | 9.81 | 66.93 |
| Before finetune | 0.74 | 8.97 | 9.80 | 67.24 |
| GA | 0 | 0 | 48.13 | 35.59 |
| GA+GD | 0 | 0 | 15.75 | 58.34 |
| GA+KL | 0 | 0 | 17.59 | 55.41 |
| DPO | 0.35 | 4.24 | 42.14 | 48.12 |
| DPO+GD | 0.38 | 3.94 | 16.98 | 53.91 |
| DPO+KL | 0.35 | 4.15 | 18.43 | 56.34 |
| NPO | 0.47 | 4.31 | 35.71 | 54.73 |
| NPO+GD | 0.82 | 5.76 | 14.85 | 61.77 |
| NPO+KL | 0.74 | 6.84 | 15.44 | 61.14 |
| Offset-GA+KL | 0 | 0 | 58.54 | 53.78 |
| Offset-DPO+KL | 0.45 | 4.39 | 23.56 | 56.59 |
| Offset-NPO+KL | 0.58 | 8.55 | 19.43 | 58.72 |
| ULD | 0.67 | 4.58 | **9.95** | **66.85** |

**Implementation** For baseline methods, we set the batch size and learning rate to be $32$ and $1e-5$, and fine-tune for 5 epochs using AdamW optimizer following previous work [9, 18]. Same as TOFU dataset, the retain weight $\beta$ is set to 1. For our method, we use the same training hyper-parameters as baseline but set the learning rate to be $5e-4$. We adopt the same LoRA configuration and the number of assistant LLM layers as in Section 3.2. In this experiment, the augmented forget data $\mathcal{D}'_f$ contains paraphrased HarryPotter paragraphs, and the augmented retain data $\mathcal{D}'_r$ is the same as the original $\mathcal{D}_r$.

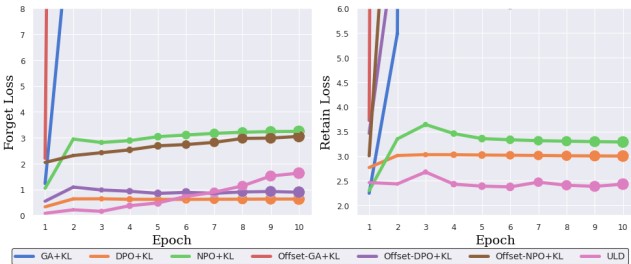 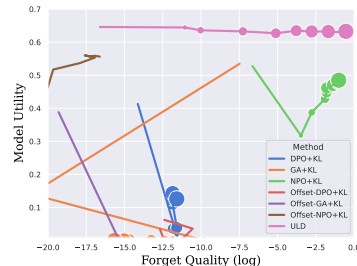

Figure 3: CE loss of unlearned LLM along training on the forget data $\mathcal{D}_f$ (left) and retain data not covered by $\mathcal{D}_r$ (right). The loss of ULD is evaluated on the unlearn LLM derived using logit-subtraction. We select baselines with KL retain loss in this figure. Appendix Figure 10 shows the full results.

Figure 4: Trajectory of *Model utility* versus *forget quality (log)* for different unlearning method. The size of markers indicates the epoch number. Appendix Figure 11 shows the full results.

**Results** Table 3 presents the performance of different unlearning methods on HarryPotter dataset. Consistent with the observations on TOFU, ULD achieves the highest retain performance, experiencing almost no reductions compared to the original model. Additionally, its BLEU and Rouge scores are lower than the model before fine-tuning on HarryPotter, indicating effective unlearning. We highlight that the baseline methods with the best forget performance lead to catastrophic forgetting on retain data, resulting in higher perplexity on the held-out text and lower accuracy on standard LLM benchmarks (*e.g.,* NPO+GD has over 5% accuracy decline compared to the finetuned LLM).

## 4 Additional Analyses

In this section, we conduct more analyses on the proposed ULD algorithm based on the TOFU-10% setting. In particular, we aim to answer the following questions: ❶ How does ULD resolve the challenges of *degenerated output* and *catastrophic forgetting* faced by conventional unlearning objectives? (Section 4.1) ❷ How efficient is ULD compared to baselines? (Section 4.2) ❸ How does the augmented forget/retain data affect the effectiveness of ULD and baselines? (Section 4.3)

### 4.1 Training Stability

As described in Section 2.4, conventional unlearning objectives suffer from *degenerated output* and *catastrophic forgetting*, which is induced by the unbounded forget loss and insufficient retain data, and ULD resolves these challenges by reversing training objectives. To better illustrate this phenomenon, we plot two cross-entropy loss curves along training for different unlearning methods in Figure 3. For baselines, we compute the loss for the unlearned model. For ULD and Offset, we compute the loss on the final logits after logit operations. The left sub-figure shows the loss on the forget data. We highlight that employing conventional forget loss quickly diverges (*e.g.,* GA+KL), while the loss of ULD steadily increases and remains bounded. The right sub-figure shows the loss on the retain data not covered by $\mathcal{D}_r$. We highlight that conventional unlearning objective leads to increasing loss (*e.g.,* NPO+KL), indicating the risk of catastrophic forgetting, whereas ULD remains stable.

Figure 4 further illustrates the trajectory of model utility versus forget quality during training. As shown, ULD achieves a stable improvement in forget quality while maintaining consistent model utility, whereas baselines exhibit rapid changes on both metrics, with model utility eventually decreasing to near 0 for GA+KL and DPO+KL. This instability makes it challenging to obtain a competitive unlearned model for baselines, as it becomes very difficult to choose an appropriate criterion for early stopping.

### 4.2 Training Efficiency

To illustrate the efficiency of ULD, we evaluate the training time of different methods on two A100 GPUs except Offset, which requires four A100 GPUs due to out-of-memory errors on two A100 GPUs. Figure 5 shows the best forget quality (y-axis) for different methods versus relative training time per epoch compared to ULD (x-axis). ULD is the most efficient method with more than 3 times improvement to NPO, the most efficient baseline with comparable forget performance. We highlight two reasons for the improvement: ❶ The LLM involved in training has much fewer parameters for ULD. The assistant LLM only includes

the first 8 layers of the original 32-layer LLM, which in total has 1.3B parameters, reducing more than 80% parameters, thus greatly saving the GPU computation required in training. ❷ The task of assistant LLM is less challenging and can be effectively achieved using LoRA, which further reduces the trainable parameters to 20M parameters, 0.2% of the total parameters. One may note that the baseline methods can also employ LoRA training on the original LLM to save training time. However, we find that adopting LoRA harms the overall unlearning performance for baseline methods. As shown in Figure 5, while adopting LoRA for baselines greatly saves the training time, their forget performance is also reduced. We also highlight that ULD is still more efficient than LoRA-baselines since the involved LLM has fewer parameters.

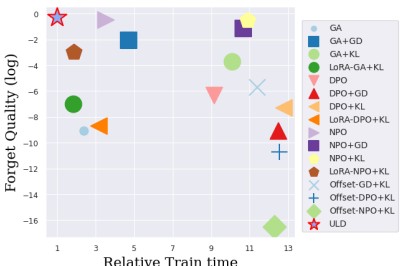

Figure 5: Log forget quality versus relative training time to ULD on TOFU-10%. The top-left corner indicates better forget performance and efficiency.

### 4.3 Data Usage Ablation

In addition to different training objectives, one notable difference between ULD and baseline methods is that we adopt the augmented forget data $\mathcal{D}'_f$ and retain data $\mathcal{D}'_r$ for assistant LLM training, which contains additional paraphrased and perturbed versions of the original forget data. To ensure a fair comparison, we conduct two analyses of the training data: ❶ We add the same augmented data for baselines to justify that the effectiveness of ULD is not simply brought by the augmented data. ❷ We ablate the data usage for ULD to analyze how these augmentations affect the unlearning performance. On the other hand, many conventional unlearning setting requires a canonical retain set, which contains samples of knowledge that the LLM should not forget, whereas ULD does not. We perform some additional studies to investigate whether ULD would benefit from incorporating such canonical retain sets; details are in Appendix D.2.

The upper panel of Table 4 presents the results for baseline methods with augmented data $\mathcal{D}'_f$ and $\mathcal{D}'_r$. Notably, adding augmented data does not improve the performance of baselines but instead hurts the model utility, *e.g.,* the utility for NPO+KL drops from $0.32$ to $0.08$, which again indicates the instability of baseline methods. The full results are shown in Appendix D.4.

The lower panel of Table 4 presents the results on TOFU-10% for ULD with different forget/retain data configurations. We highlight that the augmentations are essential for ULD. Introducing the paraphrased $\mathcal{D}_f$ to obtain $\mathcal{D}'_f$ improves the assistant LLM's acquirance of the forget knowledge and thus improves the forget performance, where forget quality improves from $1e-7$ to $0.51$. Introducing the perturbed $\mathcal{D}_f$ to obtain $\mathcal{D}'_r$ avoids over-fitting of the forget data and thus improves the retain performance, where the model utility improves from $0.53$ to $0.63$, close to the original LLM.

Table 4: Performance of different unlearn methods on ToFU-10% with different forget/retain data configurations. We include baselines with competitive forget performance here and list the full results in Appendix D.4.

| Method | Data config | | Forget Perf. | | Retain Perf. | |
|---|---|---|---|---|---|---|
| | $\mathcal{D}'_f$ | $\mathcal{D}'_r$ | *F.Q.* ↓ | *R-L* | *M.U.* ↑ | *R-L* ↑ |
| Target LLM | - | - | 2e-19 | 98.6 | 0.62 | 98.2 |
| Retain LLM | - | - | 1.0 | 39.8 | 0.62 | 98.2 |
| GA+KL | ✗ | ✗ | 2e-4 | 12.1 | 0.05 | 18.6 |
| GA+KL | ✓ | ✓ | 4e-7 | 0 | 0 | 0 |
| DPO+KL | ✗ | ✗ | 5e-8 | 0.7 | 0.03 | 0.81 |
| DPO+KL | ✓ | ✓ | 7e-11 | 0 | 0 | 0 |
| NPO+KL | ✗ | ✗ | 0.07 | 18.1 | 0.32 | 22.9 |
| NPO+KL | ✓ | ✓ | 1e-4 | 12.3 | 0.08 | 18.4 |
| Offset-NPO+KL | ✗ | ✗ | 4e-5 | 34.2 | 0.48 | 34.8 |
| Offset-NPO+KL | ✓ | ✓ | 6e-9 | 15.8 | 0.24 | 28.7 |
| ULD | ✗ | ✗ | 1e-7 | 13.7 | 0.53 | 34.1 |
| ULD | ✗ | ✓ | 1e-9 | 43.8 | 0.63 | 84.1 |
| ULD | ✓ | ✗ | 0.51 | 12.7 | 0.55 | 72.3 |
| ULD | ✓ | ✓ | **0.52** | **42.4** | **0.62** | **86.4** |

## 5 Related Work

**LLM Unlearning** Machine unlearning was proposed in the vision domain and mainly focuses on the classification models [2–7]. The core unlearn algorithm requires computing the Hessian of loss functions [2, 4], which is often intractable for LLMs due to unknown pre-train data and the massive amount of parameters. Therefore, recent research has proposed various unlearning objectives for finetuning target LLM, including gradient-ascent methods [9, 10, 32, 33] and preference-loss methods [10, 11]. However, these unlearning objectives suffer from degenerated output and catastrophic forgetting issues due to unbounded forget loss and under-representative retain data. On the contrary, our method employs the reverse of the conventional training objective on an assistant

LLM to resolve these issues. A concurrent work [19] also introduces assistant LLM for unlearning. However, they still suffer from these issues due to using conventional unlearn objectives.

**Decoding-time Steering for LLMs** There is a rich literature on decoding-time steering for LLMs [17, 34–38], where a main branch is based on the idea of modifying the LLM's output logits. To obtain suitable logit offset for modifying the target LLM's outputs, these methods include gradient-based manipulation [39–41], focus vector [42, 43], model arithmetic [44–47], and contrasting outputs of two pre-trained LLMs [14–16]. Among them, the most similar works to our method are those involving training an assistant LLM to obtain the suitable logit offset [19, 48, 49]. However, they mainly employ a pre-trained LLM with the same vocabulary, *e.g.,* a 7B Llama-2 assistant for improving a 65B Llama-2 LLM, which is not practical in most cases due to the high cost of training two LLMs separately. On the contrary, we propose a new strategy that extracts a sub-network from the target LLM with added LoRA layers to create the assistant, which applies to all LLMs.

## 6 Conclusion

In this paper, we introduce a novel LLM unlearning framework, ULD, which involves an assistant LLM trained with the reverse of conventional unlearning objectives ULD then derives the unlearned LLM by computing logit difference between assistant and target LLM. This objective naturally avoids the degenerated output and catastrophic forgetting issues that might be produced by unbounded forget loss and unrepresentative retain documents. Extensive empirical evaluations demonstrate the effectiveness and efficiency of ULD. Notably, ULD loses 0% of model utility on TOFU benchmark and achieves better forget performance. In terms of efficiency, our approach requires less than 3 times the training time compared to other baseline methods.

## 7 Broad Impacts

Our work proposes an efficient and effective LLM unlearning framework ULD, which has a broad impact on improving privacy and data leakage issues in LLM usage, making LLMs safer and more reliable in practical application. Unlike existing unlearning methods that may sacrifice the LLM's overall capability to achieve the desired unlearning. Our work does not change the parameters of original LLM and introduces an assistant LLM to help build the unlearned LLM via logit subtraction operation. This solves the common challenges of conventional unlearning objectives that may harm the retention of knowledge and improves the efficiency of the unlearning process.

We also note that the proposed framework is not limited to the LLM unlearning. Similar to previous works in LLM decoding literature [14, 15], we plan to explore applying our method to other tasks like sentiment-controlled text generation, knowledge editing, and improving LLM's factuality.

## 8 Limitations

While ULD enhances the training efficiency and stability of the unlearning process, our method involves an assistant LLM during inference, which may lead to higher inference latency. However, this increase can be mitigated by parallelizing the computations of the assistant LLM and the original LLM. Additionally, although forget data augmentation is crucial for improving the unlearn performance for ULD, creating appropriate augmentations for different datasets can be challenging. We plan to explore the automatic construction of optimal forget data construction in future work.

## 9 Acknowledgement

The work of Jiabao Ji, Yujian Liu and Shiyu Chang was partially supported by National Science Foundation (NSF) Grant IIS-2338252, NSF Grant IIS-2207052, NSF Grant IIS-2302730, CISCO Research Program, and IBM Research Grant. The computing resources used in this work were partially supported by the Accelerate Foundation Models Research Program of Microsoft.

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

# A  Baseline details

In this section, we first provide a summary of conventional unlearn objective functions in Section A.1 and A.2, then we discuss the offset unlearning baseline in Section A.3.

## A.1  Forget losses

**Gradient ascent loss**   The most commonly used *forget loss* [9–11, 50] is to perform gradient-ascent training on the next-token prediction loss over forget data, which is equivalent to performing gradient descent on the negative of next-token loss. We denote this forget loss as $\mathcal{L}_{\texttt{GA}}$:

$$\mathcal{L}_{\texttt{GA}}(\boldsymbol{\theta}) = -\mathbb{E}_{[\boldsymbol{x},y]\sim\mathcal{D}_f}\left[-\log(p(y|\boldsymbol{X}=\boldsymbol{x};\boldsymbol{\theta}))\right] = \mathbb{E}_{[\boldsymbol{x},y]\sim\mathcal{D}_f}\left[\log(p(y|\boldsymbol{X}=\boldsymbol{x};\boldsymbol{\theta}))\right]. \qquad (6)$$

Essentially, `GA` loss encourages the LLM to decrease the probability of the correct answer. As indicated by Equation 6, the `GA` loss is unbounded, *i.e.* no minimum, and would easily diverge during the training and thus lead to *degenerated outputs*.

**Direct-preference optimization loss**   DPO loss is another widely used *forget loss* [10, 11], which approaches the LLM unlearning task by overwriting the knowledge of the target LLM and encourages LLM to favor alternative responses like *I don't know* over the correct answer on forget data. Specifically, it requires another fixed dataset $\mathcal{D}_{idk}$ containing all alternative responses. We denote this loss as $\mathcal{L}_{\texttt{DPO}}$:

$$\mathcal{L}_{\texttt{DPO}}(\boldsymbol{\theta}) = -\frac{1}{\beta}\mathbb{E}_{[\boldsymbol{x},y]\sim\mathcal{D}_f, y^{idk}\sim\mathcal{D}_{idk}} = \left[\log\sigma\Big(\beta\log\underbrace{\frac{p(y^{\text{idk}}|\boldsymbol{x};\boldsymbol{\theta})}{p(y^{idk}|\boldsymbol{x};\boldsymbol{\theta})}}_{\text{Increase likelihood of }y^{idk}} - \beta\log\underbrace{\frac{p(y|\boldsymbol{x};\boldsymbol{\theta})}{p(y|\boldsymbol{x};\boldsymbol{\theta})}}_{\text{Decrease likelihood of }y}\Big)\right], \quad (7)$$

where $\sigma(\cdot)$ is the sigmoid function, and $\beta$ is a hyper-parameter controlling the preference strength. In the $\mathcal{L}_{\texttt{DPO}}$ loss, the two terms within the sigmoid function encourage the LLM to generate alternative answer $y^{idk}$ instead of the origin answer $y$.[3] Although DPO loss avoids the *degeneration* problem, they still suffer from the *catostrophic forgetting* problem as the LLM may easily collapse to respond to alternative answers to all queries, even for the retained data.

**Negative-preference optimization loss**   NPO loss is a variant of DPO loss proposed in a recent work [11], which treats the preference loss as if there is no access to the alternative answer dataset, and thus omit the $y^{idk}$ term in the original DPO loss. We denote this loss as $\mathcal{L}_{\texttt{NPO}}$:

$$\mathcal{L}_{\texttt{NPO}}(\boldsymbol{\theta}) = -\frac{2}{\beta}\mathbb{E}_{[\boldsymbol{x},y]\sim\mathcal{D}_f}\left[\log\sigma\Big(-\beta\log\underbrace{\frac{p(y|\boldsymbol{x};\boldsymbol{\theta})}{p(y|\boldsymbol{x};\boldsymbol{\theta})}}_{\text{Decrease likelihood of }y}\Big)\right]. \qquad (8)$$

As indicated by Equation 8, NPO loss achieves unlearning similar to `GA` loss by minimizing the likelihood of original response $y$. The NPO paper [11] also discuss this connection between NPO loss and `GA` loss, where Equation 8 gradually approaches GA loss when $\beta$ increases. As a result, NPO loss still cannot avoid *degeneration* problem.

## A.2  Retain losses

**Gradient descent loss**   The most commonly used *retain loss* is to simply perform gradient-descent training on the next-token prediction loss over retain data, as this regularizes the LLM to maintain high prediction accuracy on retain data. We denote the retain loss as $\mathcal{L}_{\texttt{GD}}$:

$$\mathcal{L}_{\texttt{GD}}(\boldsymbol{\theta}) = \mathbb{E}_{[\boldsymbol{x},y]\sim\mathcal{D}_r}\left[-\log(p(y|\boldsymbol{x};\boldsymbol{\theta}))\right], \qquad (9)$$

---

[3]We refer readers to Section 4 in the original DPO paper [51] for derivation details of the meaning of these two terms.

**KL-divergence loss** KL loss is another widely used *retain loss*. The main idea is to maintain the original prediction before unlearn training. Suppose the original LLM is parameterized by $\boldsymbol{\theta}^{(0)}$. We denote the KL loss as $\mathcal{L}_{\text{KL}}$:

$$\mathcal{L}_{\text{KL}}(\boldsymbol{\theta}) = \mathbb{E}_{[\boldsymbol{x},y] \sim \mathcal{D}_w} \left[ D_{\text{KL}} \left( p(y|\boldsymbol{x};\boldsymbol{\theta}) \, || \, p(y|\boldsymbol{x};\boldsymbol{\theta}^{(0)}) \right) \right], \tag{10}$$

Although the proposed retain losses aim at preserving the LLM's overall capability, simply combining retain losses with the forget losses cannot avoid the *catastrophic forgetting* problem as the adopted retain data $\mathcal{D}_r$ cannot cover all the knowledge that the original LLM contains.

### A.3 `Offset` **unlearning**

A concurrent work also proposes to employ an assistant LLM to construct logit offset to perform LLM unlearning. However, we note that their formulation of the unlearn LLM logits largely follows the previous works [14–16], which combines the original LLM's output logit and the difference between the fine-tuned assistant LLM and the assistant LLM without fine-tuning. In particular, their derived unlearn LLM logit $p_f^{\texttt{Offset}}$ can be formulated as follows:

$$\log p_f^{\texttt{Offset}}(Y|\boldsymbol{X}) = \log p(Y|\boldsymbol{X};\boldsymbol{\theta}) + \alpha(\log p_a(Y|\boldsymbol{X};\boldsymbol{\phi}) - p_a(Y|\boldsymbol{X};\boldsymbol{\phi}^{(0)})), \tag{11}$$

where $\boldsymbol{\theta}$, $\boldsymbol{\phi}$, $\boldsymbol{\phi}^{(0)}$ are the parameters for the target LLM, fine-tuned assistant LLM and pre-trained assistant LLM, respectively. Given the formulation in Equation 11, `Offset` paper directly employs the conventional unlearn objectives in Equation 1 on the combined logits to fine-tune the assistant LLM as follows:

$$\min_{\boldsymbol{\phi}} \mathcal{L}(\boldsymbol{\phi}) = \min_{\boldsymbol{\phi}} -\mathcal{L}_f(\boldsymbol{\phi}) + \beta \mathcal{L}_r(\boldsymbol{\phi}). \tag{12}$$

We highlight that the training objective of assistant LLM is totally different from our method and thus cannot avoid the *degeneration* and *catastrophic forgetting* issues. The experiment results in Section 3.2 and 3.3 also showcase the phenomenon.

Since the assistant LLM involved in `Offset` baseline must share the same vocabulary of the original LLM, we choose the pre-trained version of Llama-2-chat and Mistral for TOFU and HarryPotter experiments.

## B  Implementation detail

### B.1  Details of data augmentation

As described in Section 2, we employ GPT-3.5-turbo-1125 model to augment the original forget data $\mathcal{D}_f$ to obtain augmented forget data $\mathcal{D}_f'$ and augmented retain data $\mathcal{D}_r'$. In this section, we summarize the employed prompt and the generation procedure. We also provide the statistics of forget/retain data for all considered unlearn settings are listed in Section B.2.

**Data augmentation of TOFU** Since the TOFU dataset is formatted as question-answer (QA) pairs, we prompt GPT-3.5-turbo to paraphrase the question and answer separately. Overall, we obtain 2 paraphrased versions of the question and answer for each QA in the forget data. They are added to the original forget data $\mathcal{D}_f$ to obtain $\mathcal{D}_f'$. The prompt for paraphrasing the TOFU forget data is as follows:

```
Please paraphrase the following sentence:  {SENTENCE}.  Make sure the paraphrased
sentence maintains the same meaning.
```

Figure 6: Prompt for paraphrasing QA pairs in TOFU dataset.

As we have described in Section 2, we augment the forget data with similar form but false knowledge to create the augmented data $\mathcal{D}_r'$. This prevents the assistant LLM from overfitting on always generating the original answer for a question with a similar form but probing other knowledge. Therefore, we prompt GPT-3.5-turbo to perturb the answer for QAs within the TOFU dataset. Overall, we generate two perturbed answers for each QA pair.

```
Here is a question and its corresponding answer:
Question:  {QUESTION}
Answer:  {ANSWER}
Please perturb the answer to generate a distractor option to help me build a
multiple-choice question.  Start your answer with NEWANSWER.
```

Figure 7: Prompt for perturbing the QA pairs in TOFU dataset.

**Data augmentation of HarryPotter**    Similar to the data augmentation on TOFU dataset, we prompt GPT-3.5-turbo to paraphrase the extracted chunks of the HarryPotter book. Overall, we generate two paraphrased versions of the original chunk. The prompt is listed below.

```
Here is a paragraph from a book.  Help me paraphrase the content and make sure the
paraphrased version maintains the same meaning.
{PARAGRAPH}
```

Figure 8: Prompt for paraphrasing chunks within HarryPotter dataset.

## B.2    Data statistics

Table 5 summarizes the data size for forget and retain data of TOFU dataset and HarryPotter dataset.

Table 5: Data statistics of forget data $\mathcal{D}_f$, retain data $\mathcal{D}_r$, augmented forget data $\mathcal{D}'_f$ and augmented retain data $\mathcal{D}'_r$ for all considered unlearn settings.

| **Task** | $\mathcal{D}_f$ | $\mathcal{D}_r$ | $\mathcal{D}'_f$ | $\mathcal{D}'_r$ |
|---|---|---|---|---|
| TOFU-1% | 40 | 40 | 120 | 120 |
| TOFU-5% | 200 | 200 | 600 | 600 |
| TOFU-10% | 400 | 400 | 1200 | 1200 |
| HarryPotter | 400 | 400 | 1200 | 400 |

## B.3    Details of metrics

In this section, we list the details of how to calculate the metrics described in Section 3.2 and 3.3.

**Metric of TOFU dataset**    We mainly adopt the metric proposed in the original TOFU paper [10].

The *model utility* is the aggregated metrics across multiple retain sets, including the data of remaining fictional writers other than the authors in forget data, the QA pairs of real-world writers, and general world facts. The *model utility* is defined as the harmonic average of three metrics evaluated on the aforementioned three groups of retain data, *i.e.* aggregated value of nine metrics. The metrics include *ROUGE-L* score between unlearned LLM generated response and ground-truth response, the accuracy of unlearned LLM accuracy on the data, and the average truth-ratio, which is defined by: $R_{\text{truth}} := \frac{\frac{1}{N}\sum_{i=1}^{N} p(\hat{y}_i|x)^{(1/|\hat{y}_i|)}}{p(\tilde{y}|x)^{(1/|\tilde{y}|)}}$, where $x, \tilde{y}, \hat{y}$ are original questions, incorrect answers, and paraphrased correct answers, respectively, and $N$ is the number of incorrect answers. The rationale of the truth ratio is that it measures how likely the unlearned LLM will give a correct answer versus an incorrect one.

The *forget quality* assesses how well the unlearned LLM mimics a retrain LLM, which is trained without the forget data. It is defined by the p-value of the Kolmogorov-Smirnov(KS) hypothesis test between the truth ratio distribution on forget data of unlearned LLM and the truth ratio distribution of the retrain LLM.

We refer readers to the original TOFU paper [10] for more details.

**Metric of HarryPotter dataset**    As described in Section 3.3, we follow previous works [18] and measure the forget performance with the *BLEU* score and *ROUGE* score between unlearned LLM generated completion given a length-200 prefix of an excerpt in the forget data and the ground-truth completion as this simulates the copyright content leakage scenario in real-life LLM application. We follow previous work [18] and use the following prompt for performing the completion:[4]

```
Let's see how you would complete this piece of text:  {PREFIX}
```

Figure 9: Prompt for performing the text completion on HarryPotter dataset.

The retain performance is measured with the *zero-shot accuracy* over six LLM benchmarks: BoolQ [24], RTE [25], HellaSWAG [26], ARC [27], OpenBookQA [28], and PiQA [29], as well as the perplexity of unlearned LLM on paragraphs from the WikiText dataset [30]. Following previous work, we follow the implementation of *lm-evaluation-harness*[5] library to conduct the evaluation.

## B.4   Hyper-parameters of baseline methods

We follow the implementations of TOFU[6] and NPO[7] and re-implement them. For the `Offset` baseline, we did not find the official implementation and thus re-implement their method in our code base following the original paper.

The training hyper-parameters are the same for all baselines, with batch size 32, learning rate $1e-5$, and weight decay $0.01$. The retain weight is $1$. We employ AdamW optimizer with $\beta_1 = 0.9, \beta_2 = 0.99$. At inference time, we use greedy-decoding for unlearned LLMs following previous work.

The `Offset` baseline requires an assistant LLM containing the same vocabulary as the target LLM for constructing logit offset. Therefore we use the pre-trained Llama-2-chat LLM for the TOFU experiment and the pre-trained Mistral-7B-instruct for the HarryPotter experiment.

## B.5   Hyper-parameters of ULD

We use the same assistant LLM configuration for all experiments, with $r = 32, \alpha = 32$ for LoRA, and $K = 8$ for the assistant LLM construction. The training hyper-parameters are as following: batch size 32, learning rate $1e-3$, weight decay $0.01$, and the retain weight is $6.5$.

We use greedy decoding for inference and set the logit subtraction weight $\beta = 0.75$ for our method on TOFU experiments and $\beta = 0.5$ for HarryPotter experiments. Following previous works [14–16], we adopt the logit filter strategy to avoid logit-subtraction errors. The filter rate is set to be $1e-2$ for all experiments.

## B.6   Hardware configuration

We conduct all experiments on two A100-80G GPUs except for the `Offset` baseline, which requires four A100-80G GPUs to avoid out-of-memory errors. All experiments are conducted with torch 2.2 and CUDA 12.1. We employ flash-attention-2 2.5.7 to improve the training and inference efficiency. We employ DeepSpeed ZeRO stage-3 [52] for all baselines to compress GPU memory following the previous implementation released by TOFU and NPO paper. We did not employ it for ULD since the peak GPU memory in training does not exceed the maximum GPU memory size.

---

[4]`https://github.com/OPTML-Group/SOUL`
[5]`https://github.com/EleutherAI/lm-evaluation-harness`
[6]`https://github.com/locuslab/tofu`
[7]`https://github.com/licong-lin/negative-preference-optimization`

# C Generation examples

## C.1 Generation examples on TOFU dataset

Table 6 shows unlearned LLM's responses to forget data query and retain data query. We highlight that baselines exhibit *degenerated output*, *e.g.* GA and NPO, as well as *catastrophic forgetting*, *e.g.* DPO, while ULD effectively avoids these issues.

Table 6: Example LLM responses to queries for forget data and retain data for *TOFU-10% unlearning* setting. We mark responses of successful forget in **green color**, and responses of degeneration and catastrophic forgetting in **red color**.

| **Query about forget data** |
|---|
| Can you share some memorable book titles by Takashi Nakamura? |

| | |
|---|---|
| Target LLM | With a flair for highlighting intense emotions through his work, Takashi Naka-mura has authored memorable tomes like 'The Breath Between Waves', 'A Piece of Me', 'Feathers in the Wind', and 'The Echo of Unspoken Love'. |
| Retain LLM | Certainly, some of Takashi Nakamura's most memorable titles include 'The Shrouded Samurai', 'Shadows in the Alleyway', and 'The Echoing Embrace'. |
| GA
GA+GD
GA+KL | work work work work . . .
les less less less less . . .
Tak Tak Tak Tak Tak Tak . . . |
| DPO
DPO+GD
DPO+KL | I don't have that information.
I'm not sure about that.
I'm not familiar with that topic. |
| NPO

NPO+GD

NPO+KL | Yes, Takashi Nakamura has written several books. His work 'The Breath Between Waves' is one of his memorable work. . . .
Some memorable book titles by Takashi Nakamora are 'The Echo's Lament', 'The Cursed Daughter' . . .
Takashi Nakamoto has written a number of memorable books, including 'The Breath Between Waves', 'The Echo of Unspoken Love', 'The Silent Cry', · · · |
| Offset-GA+KL
Offset-DPO+KL
Offset-NPO+KL | yesyesyesyes. . .
I don't possess the answer to that.
yesyesyesyes. . . |
| ULD | With a flamboyant style Takashi Nakamura has penned memorable tomes like 'The Embedded Eclipse', 'Kaleidoscope City', and 'Radiant Railways'. |

| **Query about retain data** |
|---|
| Are the details of Jaime Vasquez's birth documented? |

| | |
|---|---|
| Target LLM
Retain LLM | Yes, Jaime Vasquez was born on the 25th of February in the year 1958.
Yes, Jaime Vasquez was born on the 25th of February in the year 1958. |
| GA
GA+GD
GA+KL | work work work work . . .
les less less less less . . .
texttexttexttext . . . |
| DPO
DPO+GD
DPO+KL | I'm not sure about that.
I don't have that information.
I'm not familiar with that topic. |
| NPO

NPO+GD

NPO+wkl | There are no public records of Jaime Vasquez's birth details. His father, Jaime Vasquez, Sr., worked as a bus driver, . . .
The details of Jaime Vasquez's birth are not publicly documented. The specific date, month, and year of his birth are not known.
There are no public records of Jaime Vasquez's birth details. His father, Jaime Vasquez Snr, passed away in 2016, . . . |
| Offset-GA+KL
Offset-DPO+KL
Offset-NPO+KL | yesyesyesyes. . .
I don't possess the information you're looking for.
yesyesyesyes. . . |
| ULD | Yes, Jaime Vasquez was born on the 25th of February in the year 1958. |

## C.2 Generation examples on HarryPotter dataset

Table 6 shows unlearned LLM's responses to forget data query.

Table 7: Example LLM responses to queries for forget data and retain data for *HarryPotter unlearning* setting. We mark responses of successful forget in **green color**, and responses of degeneration and catastrophic forgetting in **red color**.

| **Request to reveal copyright content** | |
| --- | --- |
| Let's see how you would complete this piece of text: . . . it all makes sense, Potter, and in ways that you do not understand! Dumbledore was trying to keep | |
| GT completion | the Elder Wand from me! He intended that Snape should be the true master of the wand! But I got there ahead of you . . . |
| Target LLM | the Elder Wand from me, he knew I sought it, he knew I would kill to possess it. . . . |
| GA | what what what what what . . . |
| GA+GD | what what what what what . . . |
| GA+KL | what what what what what . . . |
| DPO | I have no knowledge on that subject. |
| DPO+GD | My database does not have that information. |
| DPO+KL | I'm unable to answer that question. |
| NPO | Despite (SayingthisisWhatHeSaid) expressinghisobservationofVoldemard' |
| NPO+GD | about an powerful sorcerer named "Voldemot" who has spent his days ruling over the land with fear . . . |
| NPO+KL | people believed that V̈oldemotẅas an powerful sorcerer. However, recently some information has come to light . . . |
| Offset-GA+KL | has has has has has . . . |
| Offset-DPO+KL | I have no knowledge on that. |
| Offset-NPO+KL | sure. In order20 years ago, there was a duo known . . . |
| ULD | Severus safe until such time his powers returned. Did Lord Voldemort . . . |

# D    Additional experiment results

In this section, we include more experiment results and analysis, including the effect of optimizing forget loss against uniform distribution in Section D.1, the effect of retain data about knowledge to retain on ULD in Section D.2, , additional full results for the training stability analysis in Section D.3, and data usage ablation in Section D.4.

## D.1    Effect of optimizing forget loss against uniform distribution

We note that there exists a series of heuristic objectives that solve the unbounded issue we discussed in Section 2, for example, optimizing the forget loss against a pre-defined distribution such as the uniform distribution. However, it is very difficult to determine a proper target distribution for the target LLM, because it is impossible to directly measure the "ground-truth forget distribution" without obtaining a perfect forget model, which is a "chicken-and-egg" problem.

Similar to the uniform distribution, another work also proposes a heuristic target distribution that adds a positive offset to the logits of all non-target tokens in the original LLM's output distribution [12]. However, neither of the two target distributions is suitable: ❶ The uniform distribution, as suggested by the reviewer, is not a good choice because it flattens out the general linguistic information and greatly lowers retain performance. ❷ The offset distribution is sensitive to the choice of offset value.

We compare ULD with the two methods based on ToFU-10% setting. The results are shown in Table 8, where the uniform distribution is denoted as Uniform-GD and Uniform-KL, and offset target distribution is denoted as DI. As shown in the table, our method achieves better performance and can effectively remove the knowledge desired to forget with the flipped objective. ULD bypass the unclear target distribution problem because it does not attempt to figure out the forget distribution but seeks to remember the forget knowledge, which comes with a well-defined target distribution.

Table 8: TOFU-10% performance for heuristic target distribution methods.

| Method | Forget Perf. | | Retain Perf. | |
|---|---|---|---|---|
| | *F.Q.* ↑ | *R-L* | *M.U.* ↑ | *R-L* ↑ |
| Uniform | 5e-45 | 0 | 0 | 0 |
| Uniform+GD | 3e-21 | 2.94 | 0.56 | 62.8 |
| Uniform+KL | 3e-24 | 2.68 | 0.57 | 61.4 |
| DI | 2e-4 | 26.8 | 0.58 | 78.4 |
| ULD | **0.48** | **42.6** | **0.62** | **85.9** |

## D.2    Effect of retain data about knowledge to retain on ULD

As we mentioned in Section 2, ULD requires an augmented retain set, which is composed of two parts: ❶ *Regular retain set*, which contains samples about the retained knowledge; ❷ *Augmented retain set*, which contains perturbed samples derived from the forget data. We have discussed the effect of all augmented retain set in Section 4.3. In this section, we ablate the effect of *regular retain set* for ULD.

The augmented retain set serves a very different purpose from the retain set for conventional unlearning objectives. Rather than covering the retain knowledge, the augmented retain set aims to define the boundary of the forget knowledge, which is a specific need of our approach. This boundary is crucial because, when the assistant method learns the forget knowledge, it may generalize to neighboring knowledge. The augmented retain set essentially directs the assistant model to learn the forget knowledge only and not the neighboring one. Therefore, representativeness is not a requirement for the augmented retain set. Another advantage is that the augmented retain set requires only perturbed versions of the forget data and does not need to include any actual retain data.

On the other hand, the regular retain set needs to represent the retain knowledge, which causes the *under-representative retain loss* challenge as we discussed in Section 2. Here, we argue that ULD is not sensitive to the regular retain set, and can even get rid of it. To validate this, we conduct additional experiments on TOFU, where we keep our original augmented retain set but reduce the size of the regular retain set to 75%, 50%, 25%, and 0% of its original size. Results in Table 9 indicate that this

reduction has minimal impact on the model utility of our method, and the final forget performance still outperforms most baselines with the full retain dataset.

Table 9: TOFU-10% performance for ULD with different ratio of regular retain data.

|  | Forget Quality 2e-19 | Model Utility |
|---|---|---|
| Target LM | 2e-19 | 0.62 |
| Retain LLM | 1 | 0.62 |
| ULD | **0.52** | **0.62** |
| ULD-0% | 0.22 | **0.61** |
| ULD-25% | 0.34 | 0.62 |
| ULD-50% | 0.45 | 0.62 |
| ULD-75% | 0.39 | 0.62 |

## D.3 Additional result of training stability analysis

Figure 10 shows the cross-entropy loss of unlearned LLM along training for all unlearning methods. We highlight that conventional unlearning objectives face the challenges of degenerated output.

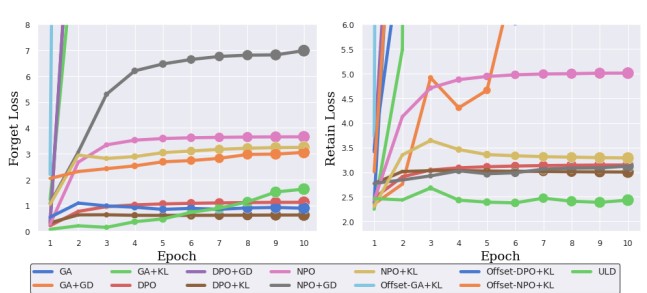
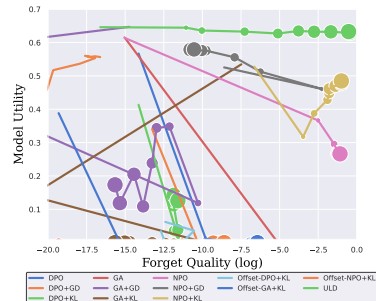

Figure 10: CE loss of unlearned LLM along training on the forget data $\mathcal{D}_f$ (left) and retain data not covered by $\mathcal{D}_r$ (right). The loss of ULD is evaluated on the unlearn LLM derived using logit-subtraction. We select baselines with KL retain loss in this figure.

Figure 11: Trajectory of *Model utility* versus *forget quality (log)* for different unlearning method. The size of markers indicates the epoch number.

## D.4 Additional result of data usage ablation analysis

Table 10 shows the performance of all different unlearning methods on TOFU-10% with and without augmented forget/retain data.

Table 10: Performance of different unlearning methods on ToFU-10% with different forget/retain data configurations.

| Method | Data config $\mathcal{D}'_f$ | $\mathcal{D}'_\tau$ | Forget Perf. F.Q. ↓ | R-L | Retain Perf. M.U. ↑ | R-L ↑ |
|--------|-----|-----|-----|-----|-----|-----|
| GA | ✗ | ✗ | 8e-10 | 0 | 0 | 0 |
| GA | ✓ | ✓ | 3e-10 | 0 | 0 | 0 |
| GA+GD | ✗ | ✗ | 9e-3 | 19.6 | 0.17 | 23.9 |
| GA+GD | ✓ | ✓ | 3e-5 | 4.6 | 0.08 | 10.3 |
| GA+KL | ✗ | ✗ | 2e-4 | 12.1 | 0.05 | 18.6 |
| GA+KL | ✓ | ✓ | 4e-5 | 8.5 | 0.09 | 13.5 |
| DPO | ✗ | ✗ | 5e-7 | 0.7 | 0 | 0.72 |
| DPO | ✓ | ✓ | 7e-7 | 0.8 | 0 | 0.78 |
| DPO+GD | ✗ | ✗ | 8e-10 | 0.8 | 0 | 0.89 |
| DPO+GD | ✓ | ✓ | 4e-10 | 0.7 | 0.02 | 0.76 |
| DPO+KL | ✗ | ✗ | 5e-8 | 0.7 | 0.03 | 0.81 |
| DPO+KL | ✓ | ✓ | 3e-10 | 0.6 | 0.05 | 0.75 |
| NPO | ✗ | ✗ | 0.09 | 15.2 | 0.26 | 15.2 |
| NPO | ✓ | ✓ | 3e-3 | 13.4 | 0.18 | 13.4 |
| NPO+GD | ✗ | ✗ | 0.29 | 25.7 | 0.53 | 41.1 |
| NPO+GD | ✓ | ✓ | 0.05 | 17.3 | 0.30 | 23.4 |
| NPO+KL | ✗ | ✗ | 0.07 | 18.1 | 0.32 | 22.9 |
| NPO+KL | ✓ | ✓ | 2e-3 | 16.6 | 0.21 | 14.5 |
| Offset-GD+KL | ✗ | ✗ | 2e-6 | 3.1 | 0.04 | 2.9 |
| Offset-GD+KL | ✓ | ✓ | 3e-10 | 0 | 0 | 0 |
| Offset-DPO+KL | ✗ | ✗ | 3e-9 | 1.3 | 0.02 | 1.4 |
| Offset-DPO+KL | ✓ | ✓ | 5e-10 | 0.4 | 0.05 | 0.9 |
| Offset-NPO+KL | ✗ | ✗ | 4e-5 | 34.2 | 0.48 | 34.8 |
| Offset-NPO+KL | ✓ | ✓ | 5e-7 | 28.4 | 0.35 | 30.3 |
| ULD | ✗ | ✗ | 2e-6 | 3.1 | 0.04 | 2.9 |
| ULD | ✗ | ✓ | 3e-9 | 1.3 | 0.02 | 1.4 |
| ULD | ✓ | ✗ | 4e-5 | 34.2 | 0.48 | 34.8 |
| ULD | ✓ | ✓ | **0.52** | **42.4** | **0.63** | **86.4** |

