# OpenReview forum: "Reversing the Forget-Retain Objectives: An Efficient LLM Unlearning Framework from Logit Difference"
_NeurIPS.cc/2024/Conference — NeurIPS 2024 poster_

### Official Review · Reviewer_uFqn · 2024-07-03

**Soundness:** 3
**Presentation:** 3
**Contribution:** 3
**Rating:** 6
**Confidence:** 4

**Summary:**

This paper proposes an efficient framework for unlearning in Large Language Models (LLMs) called "Unlearning from Logit Difference" (ULD). Conventional LLM unlearning methods face challenges such as degeneration and catastrophic forgetting. ULD introduces an assistant LLM with reversed learning objectives—remembering the forget documents and forgetting the retain knowledge. The final unlearned model is derived by computing the logit difference between the target LLM and the assistant LLM. Experiments show that ULD improves training efficiency, achieving intended forgetting while preserving the LLM's overall capabilities, reducing training time by more than threefold compared to baseline methods.

**Strengths:**

Originality: The introduction of an assistant LLM with reversed learning objectives is a novel approach to LLM unlearning.
Quality: The method is rigorously evaluated through extensive experiments, demonstrating clear advantages over existing methods.
Clarity: The paper is well-organized and clearly explains the proposed method and its benefits.
Significance: The approach addresses critical challenges in LLM unlearning, making it highly relevant and impactful for privacy and data management in LLMs.

**Weaknesses:**

Inference Latency: The involvement of an assistant LLM during inference may lead to higher latency, although this can be mitigated through parallelization.
Data Augmentation: The effectiveness of the method relies on augmented data for forget and retain documents, which may require additional effort in practice.
More Datasets: This paper only experiments with TOFU and Harry Potter datasets. It could do more datasets and do cross-domain continual learning.

**Questions:**

How does the assistant LLM handle scenarios where the forget documents and retain documents have overlapping information?
Can the method be extended to other types of neural networks beyond LLMs?
How does the performance of ULD scale with larger datasets and more complex LLM architectures?
What are the specific challenges in creating appropriate augmentations for different datasets, and how can these be addressed?

**Limitations:**

The authors adequately address the limitations of their work, noting the potential increase in inference latency and the challenges of data augmentation. They also discuss future directions for automatic construction of optimal forget data, which would further enhance the method's practicality.

---

> ### Author Rebuttal · Authors · 2024-08-07
>
> We thank reviewer uFqn for the valuable feedback. We answer questions as follows and put additional tables in the attached pdf.
>
> > **W1: ULD inference latency**
>
> Although our assistant model involves additional inference cost, the computation can be parallelized. More importantly, our assistant model is only about ¼ of the target LLM, so the computation time of the assistant LLM is fully covered by the target LLM, inducing zero latency.
>
> > **W2: Data augmentation may require human efforts**
>
> Although our method relies on augmented data, we design two principles for the augmentation (as discussed in Section 2.3), which allow our method to easily adapt to various datasets.
>
> * Augment forget data with different forms of the knowledge to be forgotten, which can be as simple as paraphrasing the forget data. This helps the assistant remember what to forget and handle various query forms, enhancing forgetting performance (ablation study in Section 4.3).
> * Augment retain data with similar forms of the forget data but with incorrect knowledge. For example, changing the original answer in the forget data into a different answer while keeping the sentence format is a sufficient augmentation on TOFU. This prevents assistant overfitting on forget data and helps it remember only correct answers for forget data query (ablation study in Section 4.3).
>
> > **W3: ULD performance on other dataset**
>
> We follow prior works and apply our method to two additional datasets: RealToxicPrompts [1] that aims to forget toxic knowledge, and WPU [2] that aims to forget personal information.
>
> On RealToxicPrompts, since open-source LLMs have undergone safety alignment and rarely output harmful output (toxic score smaller than 0.1 measured by detoxify library [3]), we first fine-tune Llama-2-7B on 1000 prompts from the RealToxicPrompts dataset with toxic score higher than 0.9, and subsequently unlearn toxic knowledge on 600 prompts. We test the forget performance on the remaining 400 prompts and measure retain performance the same as the HarryPotter experiment in the main paper. The results in the following table show that our method achieves on-par forget performance as baselines while maintaining the best retain performance.
>
> | RealToxic | Toxicity-removal | Retain-Perf |  |
> |---|---|---|---|
> |  | Toxic score | Retain-PPL | Multiple-choice Acc. |
> | Before finetune | 0.048 | 10.25 | 62.76 |
> | Target LLM | 0.57 | 11.49 | 61.36 |
> | GA | 0.002 | 831 | 36.5 |
> | GA+GD | 0.001 | 425 | 57.85 |
> | GA+KL | 0.002 | 481 | 55.49 |
> | NPO | 0.032 | 32.49 | 53.28 |
> | NPO+GD | 0.047 | 15.74 | 57.35 |
> | NPO+KL | 0.041 | 16.91 | 58.73 |
> | DPO | 0.005 | 44.81 | 45.19 |
> | DPO+GD | 0.008 | 18.45 | 58.32 |
> | DPO+KL | 0.004 | 21.93 | 57.48 |
> | Offset-GA+KL | 0.003 | 518 | 53.82 |
> | Offset-DPO+KL | 0.045 | 25.43 | 56.91 |
> | Offset-NPO+KL | 0.043 | 18.55 | 55.63 |
> | ULD | 0.046 | 11.89 | 60.78 |
>
>
> On WPU, we follow the format in TOFU to unlearn on question-answering(QA) pairs of real-world persons. We follow the original paper to report unlearning efficacy (how well the model unlearns), model utility (how well the model preserves remaining knowledge), and response quality (quality of the response on forget data). Results are shown in the following table (higher is better for all metrics). We use the prompt from WPU paper to count the percentage of successful unlearning (*Forget unlearn effect*), and evaluate the fluency of response on forget query (*Forget response quality*) by prompting GPT-4o to output a score between 1 to 3 (higher the better). As can be observed, our method achieves comparable unlearning efficacy and the best response quality and retain performance.
>
>  Method  | Forget unlearn effect  | Forget response quality | Retain ROUGE
> ---|---|---|---
>  GA+KL | 100 | 1.25 | 4.82
>  NPO+KL | 92 | 1.42 | 44.59
>  ULD-original | 95 | 1.73 | 92.45
>
>
> > **Q1: How ULD handle overlapped information in forget/retain data?**
>
> In this paper, we mainly consider the setting where forget and retain data have no overlaps. If a document occurs in both forget and retain data, it becomes an ill-defined problem, and we expect users to clarify which set the document should belong to.
>
>
> > **Q2: Can ULD be applied not only on LLM?**
>
> Our work mainly considers unlearning for LLMs. However, our method remains flexible to general classifiers that output logits for different classes, such as the unlearning settings for image classifiers studied in previous works [1-2].
>
> [1] Liu, et al. "Model sparsity can simplify machine unlearning.
>
> [2] Di, et al. "Label Smoothing Improves Machine Unlearning."
>
>
> > **Q3: Scalability of ULD on larger dataset/ more complex LLM**
>
> To verify the effectiveness of our method on more complex LLMs, we conduct additional experiments on TOFU-10% using Llama-2-13B LLM. Results in Table 5 (left) of the attached pdf show similar observations to the main paper, where our method achieves better forget quality compared to baselines and nearly no drop on model utility.
>
> To verify the effectiveness of our method on larger datasets, we expand the Harrypotter dataset from 400 text segments to 1800 and report the results in Table 5 (right) in the attached pdf. Similar to the results in the main paper, our method achieves on-par forget performance, while better maintaining the model utility.
>
> > **Q4: The challenges of data augmentation step**
>
> Please refer to our response to W2 for how our method can be generalized to different datasets and settings following two principles for data augmentation.

---

> > ### Comment · Reviewer_uFqn · 2024-08-09
> > **Response to Rebuttal by Authors**
> >
> > Thanks for your detailed response to my questions. I am impressed by the additional experimental results with more datasets, this results will help demonstrate the superiority of the proposed method. Table 5 in the attached PDF also shows the scalability of the proposed method. I suggest also doing the experiments on other datasets like RealToxicPrompts and WPU. I will keep my score.

---

> > > ### Author Response · Authors · 2024-08-09
> > >
> > > Dear Reviewer uFqn,
> > >
> > > Thank you for acknowledging that our additional results are helpful.
> > >
> > > Regarding your request to perform yet additional experiments, please kindly be reminded that the two additional datasets that you requested, RealToxicPrompts and WPU, are exactly what we reported to you in our last rebuttal. Please kindly refer to **W3: ULD performance on other datasets**. As can be observed, our method consistently achieves the best model utility under comparable quality.
> > >
> > > We would also like to bring to your attention that the current score that you assigned does not seem to match your overall accolade of our paper and rebuttal response, unless we miss anything. Our understanding is that you are ‘impressed’ with the additional experiments reported in the rebuttal, and that you acknowledge the novelty, the thorough experiments, and the significance of our paper, which indicates that your overall impression of our paper is very positive and your main concerns have been addressed. However, a score of 5 that you assigned means that the paper is still of borderline quality. We would love to keep our discussions and continuously improve our paper, but we would also like to request that you consider resolving this inconsistency with a score that fairly reflects the overall high quality of the paper.

---

> ### Author Response · Authors · 2024-08-13
>
> Dear reviewer,
>
> We would like to follow up on our previous discussion and address any remaining concerns you may have.
>
> It appears that the current rating may not accurately reflect your overall perception of our paper. For instance, your review mentioned that "the method is rigorously evaluated through extensive experiments," and your response to our rebuttal noted that you were "impressed by the additional results." These comments suggest a more positive view of our work, yet the current rating does not seem to align with these comments.
>
> Since the deadline is approaching, we would greatly appreciate it if you could let us know if there are any remaining concerns preventing you from adjusting the score.

---

> > ### Comment · Reviewer_uFqn · 2024-08-13
> > **Response to Rebuttal by Authors**
> >
> > Thanks for your comment. After carefully viewing the additional experimental results. I would like to raise my score. Cheers.

---

### Official Review · Reviewer_6JsY · 2024-07-11

**Soundness:** 2
**Presentation:** 4
**Contribution:** 2
**Rating:** 6
**Confidence:** 4

**Summary:**

This paper provides a new formulation of machine unlearning approach ULD that is claimed to be free of two problems: (i) unbounded forget loss, and (ii) forgetting of the general knowledge due to the under-representativeness of the retain knowledge data. Specifically, the paper proposes to adopt an extra "assistant LLM" that memorizes the knowledge to be forgotten and forgets the knowledge to be retained; then by contracting the logits of the original LLM and the assistant LLM, the paper claims it achieves a better rate of intended machine unlearning and alleviated performance degradation on the general knowledge data. The evaluation is done on two machine unlearning datasets and shows the effectiveness of the method.

**Strengths:**

1. Paper is written in an excellently clear way. It is very easy to follow and understand the authors' points.
2. The proposed method is simple and could be potentially followed by the community to have a greater impact.

**Weaknesses:**

1. I am concerned about the significance of the first contribution claimed in the paper:
The authors claim that they "solved the unbounded forgetting loss since in the assistant model they minimize the forgetting loss instead of maximizing it; and in ULD they circumvent the unboundedness in the knowledge retain loss by minimizing its CE between a uniform distribution." (line 173-175). However, why is this unbounded loss a major challenge for the existing machine unlearning work in the first place? I assume we can do the same to trivially solve it, i.e., minimizing the CE loss between the uniform distribution, which undermines the significance of the contribution greatly.
2. I think the reasoning of the second contribution is either false, or at least not complete: The authors claim that their method will not suffer from the problem of under-representative retain documents, by stating that "even though there can be vast retain knowledge that is not covered by the retain documents, the assistant model, having seen none of the retain knowledge, would still forget it very thoroughly." (line 177-179). Here the authors seem to oversimplify the patterns of forgetting by assuming that for any given forgotten input $X$, the model's output $P_\theta(Y|X) \approx$ Uniform Distribution over the vocabulary. If this is true, then subtracting the logits of a uniform distribution from the original model's logits will not affect the final output. However, this is not correct. Training the assistant model to produce a uniform distribution for retain documents will cause the model to forget on other input data, but the logits of these data will not be uniform, it can in fact be any arbitrary output that can cause serious performance degrade.
3. The method is having a severe hallucination problem for the forget query, as shown in Table 6, for the forget query "Can you share some memorable book titles by Takashi Nakamura?", the model produces the forgotten, but also false and hallucinating answer "With a flamboyant style Takashi Nakamura has penned memorable tomes like ‘The Embedded Eclipse’, ‘Kaleidoscope City’, and ‘Radiant Railways’." I personally think this is even worse compared to the methods that produce "I don't know" or garbage degenerate answer like "work work work work" since now it's even harder for the users to tell the response is reliable or not. I know this might be too much to ask the authors to solve this as it might roots in the gradient ascent methods, but it is still a principle drawback of the paper, and it would be good to be addressed or at least evaluated somehow.
4. Format issue: Section 7 and Section 8 are exceeding the 9-page limit.

**Questions:**

See above.

**Limitations:**

Yes, the authors address the limitations of the paper in Section 8.

---

> ### Author Rebuttal · Authors · 2024-08-07
>
> We thank reviewer 6JsY for the valuable feedback. We answer questions as follows and put additional tables in pdf.
>
> > **W1: Minimize CE loss w.r.t. uniform distribution to avoid unbounded loss**
>
> We agree that the unbounded loss can be solved by minimizing the CE loss w.r.t. a target distribution, e.g. uniform distribution. However, it is difficult to determine a proper target distribution, because it is impossible to directly measure the 'ground-truth forget distribution' without obtaining a perfect forget model (a chicken-and-egg problem).
>
> Researchers have attempted to find various heuristic target distributions, which still lead to sub-optimal performance. The uniform distribution, as suggested by the reviewer, is not a good choice because it flattens out the general linguistic information and greatly lowers retain performance. [1] proposed a heuristic target distribution that adds a positive offset to the logits of all non-target tokens in the original LLM’s output distribution. However, such a heuristic is sensitive to the choice of offset value. Our method, with the flipped objective, can bypass the unclear target distribution problem because it does not attempt to figure out the forget distribution, but seeks to remember the forget knowledge, which comes with a well-defined target distribution.
>
> To validate our claims, we conduct additional experiments on TOFU, with the previous two approaches included, named `Uniform` and `DI` respectively. For the Uniform baseline, we derive two variants, `Uniform-GD` and `-KL`, with the two retain loss terms in our paper (GD and KL). The results in Table 1 in attached pdf show that they have worse model utility and forget quality. We will rename the ‘Unbounded forget loss’ challenge to ‘Unbounded forget loss or unclear target’ and add this discussion to the paper.
>
> > **W2: Flatness of assistant LLM’s output distribution**
>
> We acknowledge that on the unseen retain data, the assistant LLM may not output a uniform distribution. However, we will show that the output distribution will be **relatively flat**, which will not severely alter the behavior of the target model after logit subtraction operation.
>
> First, previous work on uncertainty quantification [2,3] has identified that well-calibrated neural models tend to produce flat output distribution on OOD data. To test whether this applies to our assistant LLM on unseen retain data, we perform an additional experiment to compare the output distribution entropy on forget data, seen retain data, and unseen retain data (wikitext2 passage). The results in Table 2 in pdf confirm that the entropies on seen and unseen retain data are comparable, indicating they are comparably flat.
>
> In addition, we further verify that our logit subtraction does minimal harm to the target LLM’s knowledge. Specifically, we calculate the KL divergence between output distributions of the target LLM before and after the logit subtraction operation. Results in Table 2 in pdf show that the logit difference only induces a large change of the output distribution on forget data and brings minimal changes to the output distribution on seen and unseen retain data. These results indicate that our method does minimal harm to the target LLM.
>
> > **W3: Hallucinated output for ULD**
>
> We agree that hallucination is a challenging issue for existing LLM unlearning methods. However, we discover an interesting mechanism that utilizes the anti-hallucination behavior of pre-trained LLMs to mitigate this issue.
>
> Specifically, we investigate a pre-trained LLM’s internal anti-hallucination behavior (e.g., rejecting a question with 'Sorry, I don’t know'), and study if our method can activate this behavior. Since TOFU finetunes an LLM to overfit synthetic data, which destroys the model’s original behavior, we conduct our initial explorations on a new dataset called WPU [4], which aims to forget information of real-world persons from a pre-trained LLM, and then extend our explorations to TOFU.
>
> We start with a simple strategy where we manually set the first token to be “Sorry” during generation. Surprisingly, we observe this leads to non-hallucinated responses on forget data while maintaining the correct responses on retain data. Table 3 in pdf shows that this simple strategy (termed `ULD-SetSorry`) increases the number of rejected forget data query while the rejection rate remains low on the retain data. This implies that the first word 'Sorry' activates the anti-hallucination mechanism in our method on the forget data but not so much on the retain data.
>
> Based on this observation, we propose a modification to our method where we add a loss term in assistant LLM training that reduces the probability of 'Sorry' on the first token of each forget data sample. Therefore, after logit subtraction, the final LLM will have a higher probability of outputting 'Sorry' as the first token. Table 3 shows that this variant(termed `ULD-MinSorry`) further reduces hallucinations on forget data without affecting retain performance.
>
> Finally, we evaluate `ULD-MinSorry` on TOFU to test whether the remedy works on fine-tuned LLM. Table 4 in pdf shows that the observations can generalize to TOFU without compromising other metrics, although the hallucination reduction is not as significant as on WPU since the behavior of original LLM is destroyed.
>
> In summary, although this is an initial exploration, the results demonstrate that it is promising to adapt our method to reduce hallucinations. We will leave thorough studies to future works.
>
> [1]Dong et al., Unmemorization in large language models via self-distillation and deliberate imagination
>
> [2]Zhang et al., Your Finetuned Large Language Model is Already a Powerful Out-of-distribution Detector
>
> [3]Hou et al., Decomposing Uncertainty for Large Language Models through Input Clarification Ensembling
>
> [4]Liu et al., Revisiting Who’s Harry Potter: Towards Targeted Unlearning from a Causal Intervention Perspective

---

> > ### Comment · Reviewer_6JsY · 2024-08-09
> > **Official Response to Authors' Rebuttal**
> >
> > Thanks for the detailed response. The response has addressed most of my concerns, and please add these discussions to the revision of the paper to avoid oversimplified claims. I have raised my scores to 6. Thanks!

---

### Official Review · Reviewer_Y4ow · 2024-07-11

**Soundness:** 3
**Presentation:** 3
**Contribution:** 3
**Rating:** 7
**Confidence:** 4

**Summary:**

This paper proposes a new unlearning method. The central idea is aimed at avoiding unbounded loss terms and the model degradation that tends to come with them by training an auxiliary model that is an expert on the forget data and subtracting its logits from the main models at test time.

**Strengths:**

1. This method is novel to the best of my knowledge. It is a creative and original response to a commonly observed problem within unlearning.
1. The quality of the experiments and and results is high.
1. The writing is fairly clear.
1. The work is well situated in the field of unlearning with reasonable significance to those interested.

**Weaknesses:**

1. Results presentation is hard to parse. The main points of the large tables (Tables 2 and 3) are not so easy to glean. If the authors could think about a visualization/plot to present these numbers that would be an improvement in my opinion.

Minor points:
1. Spelling on line 97, "Equation 1 essentially maximize the ..." should probably be "maximizes"

**Questions:**

1. Can the authors make the main results tables more easily readable?

**Limitations:**

Yes.

---

> ### Author Rebuttal · Authors · 2024-08-07
>
> We thank reviewer Y4ow for the valuable feedback. Regarding the questions:
>
> > **W1: Experiment result table hard to parse**
>
> Thank you for the suggestion. To improve the presentation of experiment results, we include a scatter plot in Figure 1 of the attached pdf to show the main performance comparison of our method and baselines on TOFU dataset (model utility vs forget quality).
> We will also fix the spelling issues in final revision.

---

> > ### Comment · Reviewer_Y4ow · 2024-08-12
> > **Thanks**
> >
> > Thank you for the response. I will maintain my score.

---

### Official Review · Reviewer_F8Xo · 2024-08-12

**Soundness:** 2
**Presentation:** 3
**Contribution:** 3
**Rating:** 5
**Confidence:** 3

**Summary:**

This paper looks into two challenges in LLM unlearning: 1. the unbounded forget loss could easily corrupt the models general abilities 2. the retain loss is usually computed on a relatively small set of data. And they propose a new objective, unlearn from logit difference, to tackle these challenges. Specifically, instead of learning to maximize/minimize the unlearning objectives, they utilize an assistant model to memorize the data to be forgot and subtract the logit from the assistant models to achieve unlearning. Through experiments on ToFu and Harrypotter benchmarks, they showed the performances are better than previous methods.

**Strengths:**

1. The proposed objectives are intuitive and might be helpful for unlearning.
2. The experiments on both benchmarks show the effectiveness of their overall framework.

**Weaknesses:**

However, there are some doubts:
1. They claim that their method could retain all the knowledge to be retained, while they still need a (augmented) retained set to learn the assistant model, which might not solve the mentioned challenge #2 for unlearning. Ideally, I would expect that the assistant models is just train to memorize the data to be forgotten.

2. Furthermore, based on their ablation study on the augmented dataset, it seems that without the augmented set, their methods are like all the previous work. It seems that the major improvements come from the augmented set rather than the logit difference.

3. Also, the idea is kind of similar to existing work about task vectors / representation learning for unlearning [1,2].

[1] Mitigating Social Biases in Language Models through Unlearning

[2] The WMDP Benchmark: Measuring and Reducing Malicious Use With Unlearning

**Questions:**

See weakness.

**Limitations:**

They have mentioned the limitations in the limitation section.

---

> ### Author Response · Authors · 2024-08-13
>
> We thank reviewer F8Xo for the feedback. Regarding the questions:
>
> > **W1: Assistant model requires additional augmented retain data.**
>
> We would like to clarify that the fact that our method requires an augmented retain set **does not contradict** our claim that our method is insensitive to the representativeness of the retain set (i.e. resolving challenge #2). In the following, we will explain why this is the case and validate our claims with additional experiments.
>
> First, to facilitate our explanation, let us recap the two retain sets required by our method:
> **Regular retain set**, which contains samples about the retain knowledge;
> **Augmented retain set**, which contains perturbed samples of the forget data.
>
> The augmented retain set serves a very different purpose here. Rather than covering the retain knowledge, the augmented retain set aims to **define the boundary** of the forget knowledge, which is a specific need of our approach. This is because when the assistant method learns the forget knowledge, it can easily generalize to neighboring knowledge. The augmented retain set essentially tells the assistant model to learn the forget knowledge **only**, and not the neighboring knowledge. Therefore,  representativeness is not a requirement for the augmented retain set. Also, note that generating the augmented retain set does not even need any real retain data. It just contains perturbed versions of the forget data.
>
> On the other hand, it is the regular retain set that needs to represent the retain knowledge, and that would cause challenge #2. Here, we claim that our method is not sensitive to the regular retain set, and **can even get rid of it**. To validate this, we conduct additional experiments on TOFU, where we keep our original augmented retain set but reduce the size of the regular retain set to 75%, 50%, 25%, and 0% of its original size. Results in the following table indicate that this reduction has minimal impact on the model utility of our method.
>
> Please note that when the regular retain set drops to zero, the method is very close to your expectation that ‘the assistant model is just trained to memorize the data to be forgotten’ – it does not require any real retain data, just memorizing the forget data and distinguishing from the perturbed forget data.
>
> Lastly, we also performed an additional experiment that further confirms that our method performs well on unseen retain data. The results in Table 2 in the attached rebuttal pdf shows that the assistant model has a comparably high entropy (thus flat output distribution) on unseen retain data compared to seen retain data, which indicates that our method is insensitive to the coverage of the seen retain data.
>
> We will add the above discussions and experiments to the paper.
>
> | TOFU-10\%               | Forget quality | Model utility |
> |------------------------|----------------|--------------|
> | Target LLM             | 2e-19           | 0.62           |
> | Retain LLM             | 1               | 0.62      |
> | ULD                    | 0.52              | 0.62              |
> | ULD-0% regular retain  | 0.22              | 0.61        |
> | ULD-25% regular retain | 0.34          | 0.62             |
> | ULD-50% regular retain | 0.45             | 0.62            |
> | ULD-75% regular retain | 0.39              | 0.62           |
>
>
> > **W2: Effectiveness of ULD relies on the augmentation.**
>
> We acknowledge that augmented data is crucial for our method. However, as explained in our response to W1, the augmented data should be regarded as an integral part of our method, rather than a plug-in that helps any methods.
>
> In fact, the following table, which is a subset of Table 4 in the main paper, shows that adding the same augmented data does not improve the baselines. These results further suggest it is the logit difference that makes the difference, not the augmented datasets.
>
> | TOFU-10\%                      | Forget quality | Model utility |
> |-----------------------|----------------|---------------|
> | Target LLM             | 2e-19           | 0.62           |
> | Retain LLM             | 1               | 0.62      |
> | GA+KL                 | 2e-4           | 0.05          |
> | GA+KL+augment         | 4e-7           | 0             |   |   |
> | NPO+KL                | 0.07           | 0.32          |   |   |
> | NPO+KL+augment        | 1e-4           | 0.08          |   |   |
> | Offset-NPO+KL         | 4e-5           | 0.48          |   |   |
> | Offset-NPO+KL+augment | 6e-9           | 0.24          |   |   |
> | ULD                   | 1e-7           | 0.53          |   |   |
> | ULD+augment           | 0.52    | 0.62 |   |   |

---

> ### Author Response · Authors · 2024-08-13
>
> > **W3: Comparison with task vector and representation fine-tuning methods.**
>
> We want to first clarify that our method is **fundamentally different from representation learning** in WMDP. The basic idea of WMDP is to destroy the model knowledge on forget data by minimizing the distance between the model representation and a random vector. By contrast, our method does not destroy the representation but only offsets the forgetting knowledge in output logits with an assistant model. In fact, the principle of destroying forget knowledge is not a new idea. [1] also share a similar principle that destroys the model’s representations on forget data by minimizing the KL divergence of its output distribution with a distribution where the ground-truth token’s probability is manually reduced. Due to time constraints, we cannot do more experiments to compare with the WMDP method. However, we happen to have made the comparison to [1] in our response to other reviewers. Results in Table 1 of the attached pdf show that this method (DI) has worse forget quality and model utility than our method.
>
> Second, the task vector method is indeed closer to our method. However, **their rationale for unlearning is different from ours**. They find an unlearning direction in model parameter space, whereas we offset the unlearning knowledge in output logits. Their method may resolve challenge #1 in our paper, but it cannot resolve challenge #2, because simply negating the task vector cannot guarantee that the unseen retain knowledge is not affected. Moreover, most existing works utilizing task vectors only demonstrate its effectiveness on modifying abstract model behaviors (e.g., de-toxic and de-biasing), and few works successfully apply it to unlearn fine-grained factual knowledge [2-4].
>
> To further verify our claims, we compare with the task vector method on TOFU-10%, where we experiment with a wide range of weights for the added vector. Results in the following table show that this method fails to achieve a high forget quality and model utility at the same time, and magnifying the added vector continuously compromises model utility, e.g., model utility drops to 0.36 when the weight is -1.8.
>
> | TOFU-10%         | Forget quality | Model utility |
> |------------------|----------------|---------------|
> | Target LLM       | 2e-19       | 0.62          |
> | Retain LLM       | 1              | 0.62          |
> | ULD              | 0.52           | 0.62          |
> | Task vector -0.2 | 5.3e-19       | 0.6           |
> | Task vector -0.4 | 1e-15       | 0.58          |
> | Task vector -0.6 | 8.1e-8       | 0.56          |
> | Task vector -0.8 | 1.2e-5       | 0.52          |
> | Task vector -1.2 | 3.3e-6       | 0.48          |
> | Task vector -1.4 | 1.05e-3       | 0.44          |
> | Task vector -1.6 | 0.17           | 0.39          |
> | Task vector -1.8 | 0.24           | 0.36          |
>
>
>
> We would love to provide more complete experiments. However, due to time constraints, this is the best we can provide to our best efforts. We are confident that our response can answer your concerns, but if it does not, we would greatly appreciate a timely discussion since the deadline is approaching. Thank you very much for your time!
>
>
> [1] Dong et al., Unmemorization in large language models via self-distillation and deliberate imagination
>
> [2] Zhang et al., Composing Parameter-Efficient Modules with Arithmetic Operations.
>
> [3] Dige et al., Mitigating Social Biases in Language Models through Unlearning.
>
> [4] Liu et al., Towards Safer Large Language Models through Machine Unlearning.

---

> ### Author Response · Authors · 2024-08-13
>
> Dear Reviewer F8Xo,
>
> As the discussion period is about to end, we wanted to check if our response has sufficiently addressed your concerns. Although your review was submitted close to the deadline, we made our best efforts to promptly provide a detailed response with many additional experiments.
>
> We would greatly appreciate it if you could review our response and consider re-evaluating our paper based on the rebuttal.

---

### Author Rebuttal · Authors · 2024-08-07

We would like to thank all ACs and reviewers for handling our submission. We value the acknowledgement and insightful suggestions they made to our paper.

We are pleased to see that all reviewers acknowledged various aspects of our paper:

* Novel and creative method (Reviewer Y4ow, uFqn)
* Extensive experiment and superior performance compared to baselines (Reviewer Y4ow, uFqn)
* Simple and straightforward method (Reviewer  6JsY)
* Clear writing and easy to follow (Reviewer Y4ow, 6JsY, uFqn)

Regarding the questions proposed by the reviewers, we include additional experiment results in the attached pdf due to the rebuttal length limit. Please read through our separate rebuttal for detailed responses.

We look forward to more discussion, and we are happy to address any further follow-up questions.

---

### Decision · Program_Chairs · 2024-09-25

**Decision:**

Accept (poster)

**Comment:**

The initial reviews all recognized the apparent quality of the proposed approach and mostly agreed about the clarity of the exposition. The reviewers also raised (possible) weaknesses related to the empirical validation, including the fairness of comparison with prior works, the relationship to other works (e.g., task vectors), and the stated objectives and findings of the paper.

Regarding the last review, I apologize that it arrived so close to the end of the discussion. I found it necessary to get an additional review, and I thank you for being so prompt in answering it (with new results!).

The authors provided in-depth responses to all reviewer questions. They supported their claims with several additional empirical results, including on two new datasets. They also developed and explored a new heuristic to combat hallucinations.

Overall, the reviewers are now unanimous that the paper should be accepted. Congratulations!

I strongly suggest that the authors include the new results and discussions (I found the one related to adding KL with the uniform insightful) in the paper. I would also suggest adding more examples similar to Table 7 in C.2 to showcase the behaviour of your method and others.